# Benchmarking Complex Instruction-Following with Multiple Constraints Composition

**Bosi Wen**[1,†,*]   **Pei Ke**[3,*]   **Xiaotao Gu**[2]   **Lindong Wu**[2]   **Hao Huang**[2]   **Jinfeng Zhou**[1]
**Wenchuang Li**[4,†]   **Binxin Hu**[5,†]   **Wendy Gao**[2]   **Jiaxin Xu**[1]   **Yiming Liu**[1]
**Jie Tang**[1]   **Hongning Wang**[1]   **Minlie Huang**[1,‡]

[1]Tsinghua University    [2]Zhipu AI    [3]University of Electronic Science and Technology of China
[4]China University of Geosciences    [5]Central China Normal University
wbs23@mails.tsinghua.edu.cn, aihuang@tsinghua.edu.cn

## Abstract

Instruction following is one of the fundamental capabilities of large language models (LLMs). As the ability of LLMs is constantly improving, they have been increasingly applied to deal with complex human instructions in real-world scenarios. Therefore, how to evaluate the ability of complex instruction-following of LLMs has become a critical research problem. Existing benchmarks mainly focus on modeling different types of constraints in human instructions while neglecting the composition of different constraints, which is an indispensable constituent in complex instructions. To this end, we propose COMPLEXBENCH, a benchmark for comprehensively evaluating the ability of LLMs to follow complex instructions composed of multiple constraints. We propose a hierarchical taxonomy for complex instructions, including 4 constraint types, 19 constraint dimensions, and 4 composition types, and manually collect a high-quality dataset accordingly. To make the evaluation reliable, we augment LLM-based evaluators with rules to effectively verify whether generated texts can satisfy each constraint and composition. Furthermore, we obtain the final evaluation score based on the dependency structure determined by different composition types. COMPLEXBENCH identifies significant deficiencies in existing LLMs when dealing with complex instructions with multiple constraints composition[1].

## 1 Introduction

Large language models (LLMs) have proven their remarkable abilities in addressing various NLP tasks [1]. Among these, instruction following is one of the most crucial requirements for LLM applications as it determines how well LLMs align with human intents [2]. In real-world use of LLMs, almost all the tasks are formulated as instruction following, where human instructions impose different constraints on the model output to specify the requirement of specific tasks [3].

Hence, how to accurately measure the quality of instruction following has become an essential problem. While early works focused on simple and direct human instructions in traditional NLP tasks, such as translation and text classification [4, 5, 6], recent works have resorted to complex instructions consisting of multiple constraints [3, 7, 8, 9], which are important constituents of LLM's real-world use including role-play [10] and LLMs as agents [11]. These complex instruction-

---

*Equal contribution

†Work done when these authors interned at Zhipu AI.

‡Corresponding author

[1]Our dataset and codes are available at https://github.com/thu-coai/ComplexBench.

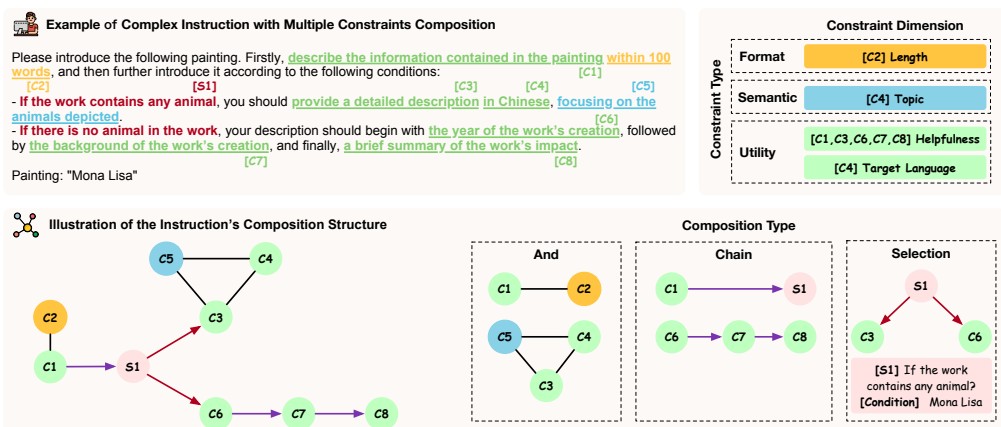

Figure 1: An example instruction of ComplexBench. All constraint dimensions contained in the instruction are marked with underlines and colors, which are categorized into three constraint types in our taxonomy: Format, Semantic, and Utility. Below is the composition structure of the instruction, where these constraint dimensions are combined through three composition types: *And*, *Chain*, and *Selection*.

following benchmarks aim to measure whether the generated text can meet every constraint in the input instruction.

However, we argue that existing complex instruction-following benchmarks neglect to model the composition of constraints, causing insufficient evaluation of the LLMs' ability to follow complex instructions. Since composition is a natural phenomenon in language use and a long-standing research problem in the NLP community [12, 13, 14, 15], it is a necessary ingredient in complex instructions to specify structural combinations of different constraints. In addition, the ignorance of composition leads to issues in both dataset construction and evaluation method design. On dataset construction, existing benchmarks are currently limited to simple composition types such as *And* which represents coordination between different constraints [3]. As shown in Figure 1, in addition to *And*, complex instructions can also include more intricate composition types of constraints, such as *Chain* (for sequential completion of constraints) and *Selection* (for conditional selection of constraints). Regarding evaluation method design, incorporating more complex composition types brings challenges in both constraint / composition evaluation and final score aggregation. First, complex instructions with structural combinations of constraints make it hard to evaluate each constraint / composition type independently with LLMs / rules due to their coupling. Then, simple aggregation methods for each constraint result, such as direct averaging, which is commonly adopted by existing benchmarks neglect the dependency among constraints brought by composition, causing potential biases in evaluation results.

In this paper, we propose ComplexBench, a novel benchmark to comprehensively evaluate the ability of LLMs to follow complex instructions. ComplexBench is manually constructed based on a hierarchical taxonomy of complex instructions, including 4 constraint types, 19 constraint dimensions, and 4 composition types, which provide a broad perspective to assess the performance of LLMs in dealing with complex instructions. To precisely measure whether LLMs' generated texts satisfy all these constraints and composition types, we design a yes / no question to verify each constraint and composition type respectively, inspired by the existing works on QA-based evaluation [16, 17, 7]. Then, we propose a new evaluation method for complex instruction-following called rule-augmented LLM-based evaluation. This method first extracts evaluation segments from generated responses for each yes / no question and then solves each question with LLMs or rules. Finally, the answers to each question are aggregated via the dependency structure among these questions, which is built based on the composition types. ComplexBench accompanied by our proposed evaluation method is expected to systematically reveal the deficiencies of existing LLMs on complex instructions and provide insights on the improvement of LLMs when dealing with various constraints and compositions. Our main contributions are as follows:

- We propose a comprehensive hierarchical taxonomy for complex instructions, including 4 constraint types, 19 constraint dimensions, and 4 composition types. We manually collect

| Benchmark | Data Size | Constraint Taxonomy | Composition Type | | | | Evaluation Method | | |
|---|---|---|---|---|---|---|---|---|---|
| | | | *And* | *Chain* | *Selection* | *Nested.* | LLM-based | Rule-based | Aggregation Function |
| WizardLM Testset [18] | 218 | - | ✔ | - | - | - | ✔ | - | - |
| CELLO [8] | 523 | 4 | ✔ | ✔ | - | - | - | ✔ | Average |
| FollowBench [3] | 820 | 5 | ✔ | - | - | - | ✔ | ✔ | Average |
| IFEval [19] | 541 | 25 | ✔ | - | - | - | - | ✔ | Average |
| InfoBench [7] | 500 | 5 | ✔ | - | - | - | ✔ | - | Average |
| CoI Testset [20] | 1,068 | - | - | ✔ | - | - | - | ✔ | - |
| COMPLEXBENCH (ours) | 1,150 | 4-19 | ✔ | ✔ | ✔ | ✔ | ✔ | ✔ | Dependency-based Aggregation |

Table 1: Comparisons between COMPLEXBENCH and other benchmarks, illustrating the features including dataset sizes, constraint taxonomies, composition types, and evaluation methods. - in Aggregation Function means there is no step to evaluate each constraint and aggregate the final score.

    a high-quality benchmark dataset for complex-instruction following, covering all types of constraints and compositions in our taxonomy.

- We accompany the benchmark with a new automated evaluation method to accurately evaluate the ability of LLMs to follow complex instructions , which integrates the advantages of LLM-based and rule-based methods to verify each constraint and composition type and aggregates the final score via the dependency structure brought by composition types.

- We conduct experiments on the proposed benchmark for a wide range of established LLMs, systematically revealing their deficiencies on various constraints and compositions.

## 2 Related Work

**Evaluation of Instruction-Following.** Instruction following remains one of the most important factors determining the practicality of LLMs [21]. Therefore, numerous studies have attempted to evaluate it from various aspects. Earlier works used to focus on simple human instructions formed with mostly a single constraint, such as semantic [5, 4, 6] and format [19, 22, 23] constraints. Since LLMs have been gradually applied to address complex real-world tasks, users have to form complex instructions, which naturally call for the evaluation of the LLMs' ability in complex instruction following [3, 7]. WizardLM [18] employs two strategies, *In-Breadth Evolving* and *In-depth Evolving*, to form complex instructions from simple ones. CELLO [8] defines complex instructions from task descriptions and input text, and evaluates LLMs with real-world scenarios data. Unlike our work, which includes subjective and objective constraints and combines LLM-based and rule-based evaluations, CELLO focuses only on objective, rule-verifiable constraints and uses rule-based scoring functions for evaluation. Nonetheless, we argue that these benchmarks neglect to model the composition of constraints, which is an important character in complex instructions and brings non-negligible structural complexity that is crucial to assessing LLMs' abilities.

**Compositionality in NLP.** Previous studies have explored compositionality across traditional NLP tasks, including semantic parsing [24, 25, 26], machine translation [26, 27], style transfer [28], and data-to-text generation [29]. However, in the task of instruction-following, how the LLMs deal with the compositionality in instructions is still under-explored. CompMCTG [30] investigates the compositionality of multiple control attributes for LLMs, which is a topic neighboring ours. Nevertheless, our work studies more complex composition types beyond simple coordination between different constraints, such as *Chain* and *Selection* and their nested structures, which form the basis of many real-world complex tasks for LLMs.

## 3 COMPLEXBENCH Framework

### 3.1 Overview

To comprehensively evaluate the ability of LLMs to follow complex instructions, we propose a hierarchical taxonomy to define constraints and composition types. For constraints, we extend common constraints in controlled text generation tasks to the instruction-following tasks and consider a two-level structure including coarse-grained types and fine-grained dimensions (Section 3.2). As for compositions that indicate structural combinations of constraints, we consider the characteristics of instruction-following tasks to define the composition types according to existing works on compositionality in traditional NLP tasks (Section 3.3).

| Composition Type | Description | Example | Illustration |
|---|---|---|---|
| *Single* | The output is required to satisfy a single constraint. | **Please summarize the following news.** | $c_1$ |
| *And* | The output is required to satisfy multiple constraints simultaneously. | **Please summarize the following news.** The summary should **be output in bullet points**, and **within 100 words**. | $c_1$ $c_2$ $c_3$ |
| *Chain* | The output is required to complete multiple tasks sequentially, each of which needs to satisfy its own constraints. | Please introduce "Mona Lisa" briefly. **Firstly,** introduce the year of creation, **then** describe the background of the work's creation, **and finally,** summarize the impact of the work. | $T_1$ $T_2$ $T_3$ |
| *Selection* | The output is required to select different branches according to certain conditions, fulfilling the constraints of the corresponding branch. | Please introduce the following painting.
- **If the work contains any animal**, the description should be in English
- **Otherwise**, the description should be in Chinese

Painting: "Mona Lisa" | $S_1(cond_1)$
$B_1$ $B_2$ |
| *Nested Structure* | The above composition types are recursively nested to form more complex structures. | Analy the sentiment of above user comment and complete the following tasks:

1. **If it's positive**:
- Identify the products within the comments …
2. **If it's negative**, analyze the reasons for it:
- **If the reason is not about products itself**, …
- **Otherwise**, … | $S_1(cond_1)$
$B_1$ $B_2$ $S_2(cond_2)$
$B_{21}$ $B_{22}$ |

Figure 3: Composition types in CoMPLEXBENCH. Each node is a part of an instruction. The purple node may contain other composition types, while the blue node does not. In addition to 4 basic types, the last row also shows a nested selection type.

## 3.2 Constraint

Following existing works on controlled text generation and instruction following [31, 32, 33, 34, 19, 10], we propose a two-level structure for constraints including 4 constraint types (i.e., Lexical, Format, Semantic, and Utility) and 19 specific constraint dimensions which are further divided from the above types. The distribution of these constraint types and dimensions within CoMPLEXBENCH is shown in Figure 2. We present the definitions of constraint types in the following and describe the details of the constraint dimensions in Appendix D.

**Lexical Constraint** requires to output specific keywords or phrases or precisely generate texts that are related to specific keywords mentioned in the instructions [35, 36, 34].

**Format Constraint** specifies the requirements on the output structure (such as JSON, Markdown, and bullet points), length, and patterns of the output, where the patterns include punctuation, content at the beginning or end, and the output templates. Format constraints require LLMs to possess a precise understanding and planning of the output content, which remain challenging for current LLMs [19, 23].

**Semantic Constraint** specifies the topic [37], language style [32], personality [10], and sentiment [38] of the output, which are common constraints in the existing works on controlled text generation.

**Utility Constraint** measures the language, helpfulness, supportiveness, consistency, and factuality of generated texts, which are holistic properties. Among these, helpfulness indicates whether the generated text can complete the basic task included in the instruction (such as *Please introduce the following painting.* in Figure 1) regardless of satisfaction of other constraints, while supportiveness means whether

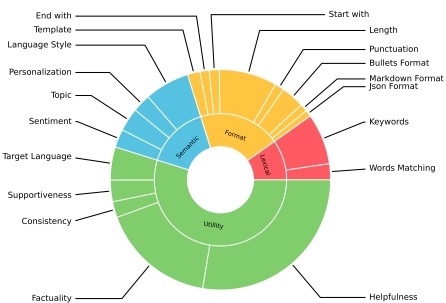

Figure 2: Constraint distribution of CoM-PLEXBENCH. The Utility constraints helpfulness and factuality possess a high proportion due to their prevalence in various instructions, which are basic requirements for high-quality outputs.

the generated text is faithful to the instruction.

## 3.3 Composition

As shown in Figure 3, we propose 4 composition types that indicate typical structural combinations of constraints.

**Single.** The output is required to satisfy a single constraint, with no composition involved.

**And.** The output needs to satisfy multiple constraints simultaneously. This simple composition type commonly appears in most of the existing benchmarks on complex instruction-following [3, 19, 7].

**Chain.** The output is required to complete multiple tasks in the instruction sequentially, each of which may contain several constraints. Formally, *Chain* contains $n$ tasks $\{T_1, T_2, \ldots, T_n\}$, which need to be completed sequentially. The output of $T_{k+1}$ may depends on that of $T_k$ ($k = 1, 2, \cdots, n-1$).

**Selection.** The output is required to select different branches according to certain conditions, fulfilling the constraints of the corresponding branch. Formally, *Selection* contains $m$ branches $\{B_1, B_2, \ldots, B_m\}$, each of which is a task with expected outputs $Y_1, Y_2, \ldots, Y_m$ respectively. We denote a selection function as $S$ with a range $\{1, 2, \cdots, m\}$, taking the selection condition *cond* as input. Finally, the expected output of the instruction is $Y_{S(cond)}$.

It's worth noting that the above composition types can be nested to construct more complex structures. Each task in *Chain* and each branch in *Selection* may also contain other composition types. As shown in the last row of Figure 3, a branch of *Selection* can also contain *Selection*, thus forming a nested selection composition type.

To verify the necessity and comprehensiveness of the composition types considered in CᴏᴍᴘʟᴇxBᴇɴᴄʜ, we analyze the distribution of composition types in real-world scenarios. We collect instructions with high demand and representativeness from an online LLM-based chat service platform that serves more than a million users daily including general and professional instructions. General instructions refer to the instructions used by individual users in routine scenarios, while professional instructions refer to those used by enterprise-level users in business and research scenarios. For each category of instructions, we randomly sample 300 instructions and manually count the number of instructions containing each composition

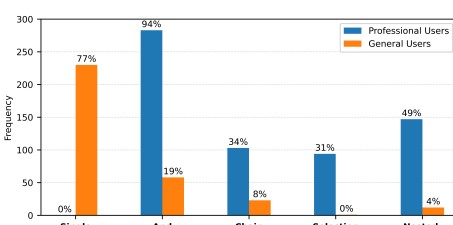

Figure 4: Composition type distribution of general and professional instructions.

type. We found that the taxonomy of CᴏᴍᴘʟᴇxBᴇɴᴄʜ fully covers present composition types. As shown in Figure 4, although the composition types of general instructions are relatively simple and have already been covered by current benchmarks, professional instructions include more complex composition types, such as *Selection* and nested structures of multiple composition types, which have rarely been considered by current benchmarks. As LLMs have been gradually applied to deal with complex instructions in professional scenarios, it is necessary to evaluate their ability to follow instructions with multiple constraints composition.

## 4 CᴏᴍᴘʟᴇxBᴇɴᴄʜ Construction

### 4.1 Data Collection

We manually construct CᴏᴍᴘʟᴇxBᴇɴᴄʜ based on the taxonomy described in Section 3. The detailed construction pipeline consists of four steps, i.e., **Reference Instructions Collection**, **Task Allocation**, **Data Annotation and Validation**, and **Selection Branch Expansion**. We initially used our proposed method to construct Chinese data, while also providing an English version of CᴏᴍᴘʟᴇxBᴇɴᴄʜ. More details are in Appendix F, G, and H.

**Reference Instruction Collection.** Considering the difficulty of constructing complex instructions from scratch, annotators are required to create new complex instructions based on provided reference instructions. We collect reference instructions from real-world application scenarios and open-source instruction following benchmarks [19, 3, 7]. We conduct strict desensitization of privacy and carefully filter these instructions using category and quality classifiers.

**Task Allocation.** To ensure comprehensive coverage of each constraint and composition type, we partition the entire dataset construction into multiple annotation tasks. Each annotation task has different requirements for the minimal number of constraint dimensions in each constraint type and composition type. Annotators are required to modify reference instructions to meet the requirements of corresponding tasks. To alleviate the annotation cost, especially when the constraint

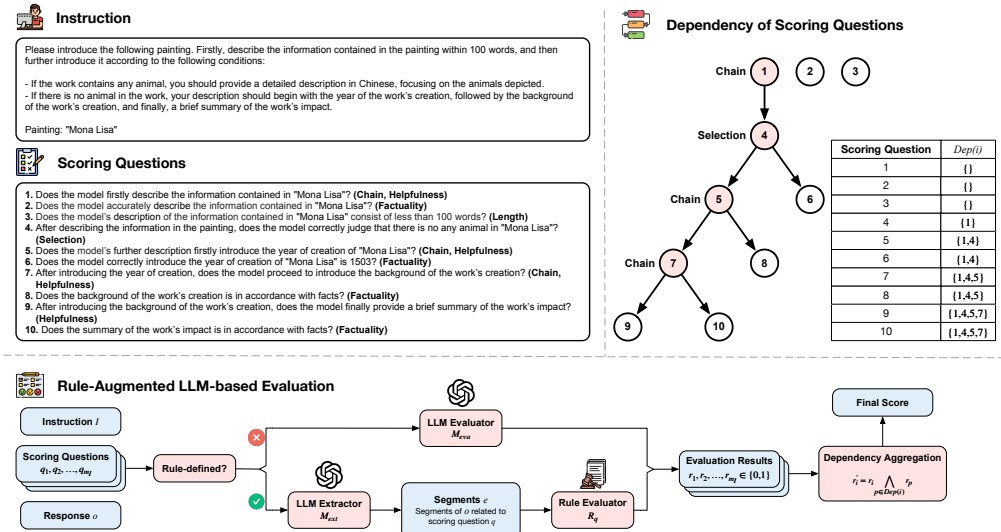

Figure 5: An exemplar evaluation process of COMPLEXBENCH. Given an instruction and its scoring questions, COMPLEXBENCH integrates the rule and LLM evaluator to verify each of them and aggregates the final score based on the dependency structure of composition types in the instruction.

dimensions in the reference instructions and task requirements are different, we leverage GPT-4 [39] to automatically acquire the constraint dimensions in reference instructions and assign them to corresponding annotation tasks according to minimal editing distance.

**Data Annotation and Validation.** Given reference instructions and corresponding annotation task requirements, annotators are expected to construct new complex instructions and annotate the constraint dimensions and composition types. After the data annotation, newly constructed instructions are cross-validated by other annotators. The process of validation continues until constructed instructions meet the following criteria: (1) Clarity & Reasonableness: The instruction should be easy to understand, unambiguous, and realistic, with at least one reasonable answer. (2) Validity of Constraints: Every constraint within the instruction should substantially influence the output. (3) Complexity & Difficulty: The instruction should be challenging for most LLMs and be capable of distinguishing the complex instruction-following abilities of different LLMs.

**Selection Branch Expansion.** When evaluating the ability of LLMs to follow instructions containing *Selection*, the predisposition toward random selection by LLMs may bring potential bias because most instructions cover only one correct selection branch. But the probability of the correct branch appearing at each position in *Selection* of data constructed by annotators is unequal [2]. To address this issue, in the final stage of instruction construction, we manually modify the selection condition based on the selection function to construct multiple instructions that cover different correct branches, ensuring an equal probability of the correct branch appearing at each position.

## 4.2 Evaluation Protocol

To conduct a detailed evaluation of how well each constraint and composition type is satisfied, we draw inspiration from previous works that transform text evaluation into multiple question-answering tasks [16, 17, 7]. For each constraint and composition type specified in an instruction, we manually craft a scoring question that can be succinctly answered with either "yes" or "no."

Current mainstream evaluation methods contain LLM-based [18, 3, 7] and rule-based methods [8, 19, 20]. In our preliminary experiments, we find that LLM-based methods are effective at answering open-ended scoring questions, but they demonstrate a significant deficiency in those involving numerical computation, counting, and other objective rule-defined areas, such as keyword inclusion and text length. Simultaneously, rule-based methods perform well in rule-defined areas but are powerless against open-ended scoring problems. To address their limitations, we design a Rule-Augmented LLM-based (RAL) evaluation method to equip LLM evaluators with rules to answer

---

[2]For instance, after manual inspection we find that in all the selection compositions with two branches, annotators have about a 70% probability of selecting the first branch as the correct one.

scoring questions in both rule-defined and open-ended areas. For the instruction $I$, the generated response to be evaluated $o$, and the scoring problem $q$, if $q$ is verifiable by rules, we first use the LLM to automatically extract segments $e$ of $o$, which is related to scoring question $q$. Subsequently, we use the rule $R_q$ written for $q$ to obtain the evaluation result $r_q \in \{0, 1\}$, that is:

$$e = \mathcal{M}_{ext}(I, q, o) \tag{1}$$

$$r_q = R_q(e) \tag{2}$$

where $\mathcal{M}_{ext}$ indicates the LLM with the prompt used for extraction. Otherwise, if $q$ cannot be verified by rules, we directly use the LLM to measure the quality of $o$:

$$r_q = \mathcal{M}_{eva}(I, q, o) \tag{3}$$

where $\mathcal{M}_{eva}$ denotes the LLM with the prompt used for evaluation. For composition types, considering that their satisfaction is a prerequisite for satisfying some constraints, we model the dependencies of its scoring questions. Specifically, for *Chain*, all the scoring questions of the subsequent task depend on the answers to those of the preceding task. And for *Selection*, all the scoring questions of the selection branch depend on whether the correct selection branch is selected. If a scoring question is judged as "no", all the scoring questions depending on it will also be directly judged as "no". Formally, we denote the set of scoring questions that $q$ depends on as $Dep(q)$. After all scoring questions have been independently verified, *Dependency Aggregation* will be performed, and the result of $q$ will be calculated as follows:

$$r'_q = r_q \bigwedge_{p \in Dep(q)} r_p \tag{4}$$

Finally, following InfoBench [7], we calculate Decomposed Requirements Following Ratio (DRFR) as the final score during *Score Aggregation*. Considering a benchmark dataset has $N$ instructions, the instruction $i$ has $m_i$ scoring questions, and the result of the $j$-th scoring question is $r'_{ij}$, the metric is calculated as: $DRFR = \sum_{i,j} r'_{ij} / \sum_i m_i$. Figure 5 shows a framework of our evaluation protocol.

### 4.3 Benchmark Statistics

 COMPLEXBENCH contains 1,150 instructions and 5,306 scoring questions, as shown in Table 2. Nesting depth means the maximum depth of composition types. In addition to three basic composition types including *And*, *Chain*, and *Selection*, we adopt a separate category whose instructions simultaneously contain *Chain* and *Selection*, aiming to use these two challenging types to explore the boundary of LLMs' ability in complex instruction-following[3]. We also present the task distribution of COMPLEXBENCH in Appendix C.

| Category | Nesting Depth | #Inst. | #Len. | #Ques. | #Con. |
|---|---|---|---|---|---|
| And | 1 | 475 | 279.39 | 4.09 | 4.14 |
| Chain | 1 | 70 | 352.11 | 4.83 | 4.94 |
| | 2 | 170 | 486.84 | 6.24 | 6.32 |
| Selection | 1 | 80 | 753.15 | 2.91 | 2.06 |
| | 2 | 224 | 664.13 | 4.40 | 3.09 |
| | ≥ 3 | 46 | 1409.93 | 5.76 | 3.78 |
| Selection & Chain | 2 | 30 | 440.37 | 4.37 | 3.63 |
| | ≥ 3 | 55 | 398.82 | 6.18 | 5.27 |
| Overall | - | 1150 | 477.51 | 4.61 | 4.19 |

Table 2: Statistics of COMPLEXBENCH including the number of instructions (**#Inst.**), the average number of characters (**#Len.**), scoring questions (**#Ques.**), and constraints (**#Con.**) per instruction.

## 5 Experiments

### 5.1 Agreement Evaluation

To measure the agreement between our evaluation method and manual evaluation, we randomly sample 200 instructions from COMPLEXBENCH to construct a meta-evaluation dataset. Five LLMs are involved in this evaluation as generation models. We employ GPT-4-1106 [39] as our primary judge and adopt two metrics to confirm the reliability of our method: (1) **Overall Pairwise Agreement**: Given an instruction, two model responses (denoted as A and B), the human annotators are instructed to compare the quality and choose from 3 options, namely A better than B, tie, B better than A. Subsequently, the automatic evaluation scores for two model responses are converted into pairwise

---

[3]Since *And* commonly appears in various instructions, we simply categorize instructions containing both *Chain* / *Selection* and *And* together with those only containing *Chain* / *Selection* into one category.

comparisons to measure agreement with human annotators. (2) **Question-level Agreement**: Given an instruction and a model response, human annotators are instructed to judge whether each scoring question is satisfied respectively. Then, we calculate the agreement between automatic evaluation results and human-annotated ones.

For the Overall Pairwise Agreement, we sample 500 pairs from the outputs of 5 LLMs. Direct Scoring serves as a baseline, which adopts a scoring prompt [5] to assign a score to the response with a scale of 1-10. As shown in Table 3, our method can improve the agreement with manual evaluations compared to Direct Scoring with a large margin. *Dependency Aggregation* also shows its important contribution to our method due to its modeling of composition structures.

| Evaluation Method | Pairwise Agreement |
|---|---|
| Ours | **0.614** |
| Ours w/o *Dep.* | 0.574 |
| Direct Scoring | 0.512 |

Table 3: Overall Pairwise Agreement with human. *Dep.* means *Dependency Aggregation*.

For the Question-level Agreement, the scoring questions in the meta-evaluation dataset are categorized into two types: (1) Rule-defined, which can be verified by rules and constitutes 17% of the total, and (2) Open-ended, which is not verifiable by rules. We compare our method with Direct Scoring, which considers a response with a score above 5 to satisfy all scoring questions of an instruction. We also remove rule arguments (w/o rule) to verify its effectiveness. As shown in Table 4, RAL outperforms all the baselines and exhibits an impressive 87.82% agreement with humans at the overall level. The LLM-based evaluator (i.e., RAL w/o rule in Table 4) shows its weakness in rule-defined areas that rule arguments mainly contribute to, supporting our motivation.

| Subset | Evaluator | Agreement between human |
|---|---|---|
| Rule-defined | RAL | **95.36%** |
| | RAL w/o rule | 82.02% |
| | Direct Scoring | 62.02% |
| Open-ended | RAL | **86.28%** |
| | RAL w/o rule | **86.28%** |
| | Direct Scoring | 77.83% |
| Overall | RAL | **87.82%** |
| | RAL w/o rule | 85.56% |
| | Direct Scoring | 75.18% |

Table 4: Question-level Agreement with human.

## 5.2 Automatic Evaluation

### 5.2.1 Setup

We use GPT-4-1106 [39] as our judge to evaluate 15 LLMs: (1) **Closed-source LLMs**: GPT-4-1106, Claude-3-Opus [40], GLM-4 [41], ERNIEBot-4, GPT-3.5-Turbo-1106. (2) **Open-source LLMs**: Qwen1.5-Chat [42], Llama3-Instruct [43], InternLM2-Chat [44], Baichuan2-Chat [45], Mistral-Instruct [46], InternLM2-Chat [44], ChatGLM3-Chat [47]. The sizes of these models vary from 6B to 72B. We use greedy search for reproducibility, and the maximum generation length is 8,192.

### 5.2.2 Main Results

The main results are shown in Table 5. **Firstly**, the widely recognized powerful GPT-4 still fails to complete 20% of complex instructions, highlighting the necessity of complex instruction evaluation. **Secondly**, as the complexity of composition types within instruction increases , the performance of all LLMs significantly drops, especially on *Selection* and *Chain*. This aligns with our motivation for constructing complex composition types. **Thirdly**, the performance of most open-source LLMs falls short compared to closed-source LLMs especially on complex composition types, indicating that open-source LLMs still have a large room for improvement in chasing the capabilities of closed-source LLMs.

To dissect the ability of LLMs to follow specific constraint and composition types, we calculate the average accuracy of scoring questions for

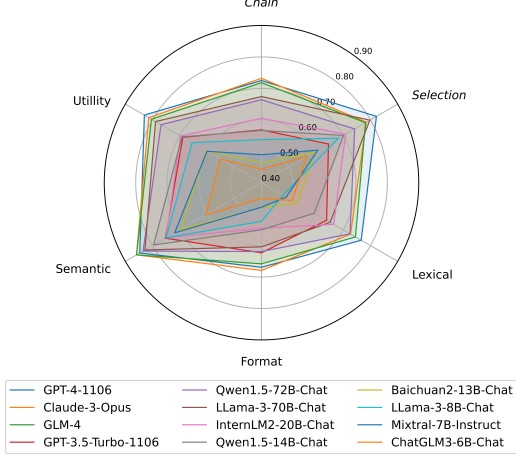

Figure 6: The performance of LLMs on different constraint and composition types.

| Category | And | Chain | | | Selection | | | | Selection & Chain | | | All |
|---|---|---|---|---|---|---|---|---|---|---|---|---|
| Nesting Depth | 1 | 1 | 2 | Avg. | 1 | 2 | ≥ 3 | Avg. | 2 | ≥ 3 | Avg. | Avg. |
| *Closed-Source Language Models* | | | | | | | | | | | | |
| GPT-4-1106 | 0.881 | **0.787** | 0.759 | 0.766 | **0.815** | **0.772** | **0.694** | **0.765** | 0.802 | 0.626 | 0.675 | **0.800** |
| Claude-3-Opus | **0.886** | 0.784 | **0.779** | **0.780** | 0.764 | 0.749 | 0.592 | 0.724 | 0.695 | 0.576 | 0.609 | 0.788 |
| GLM-4 | 0.868 | 0.763 | 0.739 | 0.745 | 0.768 | 0.739 | 0.626 | 0.724 | **0.809** | **0.647** | **0.692** | 0.779 |
| ERNIEBot-4 | 0.866 | 0.749 | 0.735 | 0.738 | 0.725 | 0.696 | 0.649 | 0.692 | 0.756 | 0.600 | 0.643 | 0.764 |
| GPT-3.5-Turbo-1106 | 0.845 | 0.686 | 0.630 | 0.644 | 0.661 | 0.561 | 0.475 | 0.561 | 0.565 | 0.482 | 0.505 | 0.682 |
| *Open-Source Language Models* | | | | | | | | | | | | |
| Qwen1.5-72B-Chat | 0.873 | 0.749 | 0.730 | 0.735 | 0.751 | 0.698 | 0.521 | 0.675 | 0.611 | 0.521 | 0.546 | 0.752 |
| Llama-3-70B-Instruct | 0.858 | 0.769 | 0.722 | 0.733 | 0.747 | 0.704 | 0.675 | 0.706 | 0.573 | 0.571 | 0.571 | 0.757 |
| InternLM2-20B-Chat | 0.796 | 0.666 | 0.648 | 0.652 | 0.648 | 0.599 | 0.543 | 0.597 | 0.611 | 0.488 | 0.522 | 0.678 |
| Qwen1.5-14B-Chat | 0.817 | 0.657 | 0.636 | 0.641 | 0.622 | 0.621 | 0.536 | 0.606 | 0.550 | 0.435 | 0.467 | 0.680 |
| Baichuan2-13B-Chat | 0.760 | 0.583 | 0.517 | 0.533 | 0.571 | 0.479 | 0.404 | 0.480 | 0.443 | 0.409 | 0.418 | 0.591 |
| Llama-3-8B-Instruct | 0.778 | 0.669 | 0.568 | 0.592 | 0.597 | 0.552 | 0.483 | 0.546 | 0.626 | 0.429 | 0.484 | 0.638 |
| Mistral-7B-Instruct | 0.737 | 0.574 | 0.556 | 0.560 | 0.554 | 0.493 | 0.411 | 0.488 | 0.534 | 0.374 | 0.418 | 0.592 |
| Qwen1.5-7B-Chat | 0.802 | 0.598 | 0.611 | 0.608 | 0.519 | 0.564 | 0.570 | 0.558 | 0.634 | 0.491 | 0.531 | 0.658 |
| InternLM2-7B-Chat | 0.755 | 0.633 | 0.598 | 0.607 | 0.532 | 0.568 | 0.525 | 0.555 | 0.550 | 0.432 | 0.465 | 0.634 |
| ChatGLM3-6B-Chat | 0.701 | 0.556 | 0.490 | 0.506 | 0.455 | 0.430 | 0.411 | 0.431 | 0.573 | 0.312 | 0.384 | 0.546 |

Table 5: DRFR of LLMs computed by our proposed RAL method. The highest performance among open-source models is underlined, while the highest performance overall is **bold**.

each type. The results are shown in Figure 6. **Firstly**, for constraints, LLMs generally perform better on Semantic and Utility constraints but struggle with the Format and Lexical constraints that have explicit evaluation standards. **Secondly**, for compositions, *Chain* presents severe challenges while *Selection* come second. We speculate that the main difficulty in *Selection* lies not only in choosing the correct branch but in executing it without interference from irrelevant branches. More results and analyses are in Appendix I.

### 5.2.3 Analysis

**Decomposition of instructions with composition types.** To explore whether decomposing complex instructions and executing them through multi-round interactions can improve the performance of LLMs, we manually decompose COMPLEXBENCH instructions based on composition types (e.g., *Chain* into sequential tasks, *Selection* into selection and execution branches, while *And* remains intact) and compare the performance of LLMs between executing decomposed instructions step-by-step and original instructions in one step. The scoring questions of original instructions are split into corresponding decomposed ones with the same dependencies to ensure a fair comparison.

Table 6 shows that GPT-3.5-Turbo-1106 generally performs worse in decomposed instructions, especially as the complexity of composition types within instructions increases. We conjecture that this is due to cumulative errors in multi-round interactions, highlighting that our benchmark is challenging and cannot be simply solved via instruction decomposition.

**The Coherent Test for Selection.** To comprehensively measure the performance of LLMs on different conditions of *Selection*, we merge the instructions with the same branches and selection functions but different conditions into the same task group. For example, the instruction about the Mona Lisa shown in Table 1 and an-

| Category | Nesting Depth | Origin | Decomposition | Δ |
|---|---|---|---|---|
| **And** | 1 | 0.845 | 0.845 | 0.000 |
| **Chain** | 1 | 0.686 | 0.655 | -0.031 |
| | 2 | 0.630 | 0.583 | **-0.047** |
| **Selection** | 1 | 0.661 | 0.631 | -0.030 |
| | 2 | 0.561 | 0.520 | -0.041 |
| | ≥ 3 | 0.475 | 0.411 | **-0.064** |
| **Selection & Chain** | 2 | 0.565 | 0.504 | -0.061 |
| | ≥ 3 | 0.482 | 0.415 | **-0.067** |
| **Overall** | - | 0.682 | 0.652 | -0.030 |

Table 6: The performance of GPT-3.5-Turbo-1106 on original and decomposed instructions.

other instruction where everything else remains the same except the final condition "Painting: Mona Lisa" is changed to "Painting: Galloping horse" are merged into the same task group. The two instructions need to execute two different selection branches. We calculate the proportion of instructions with all scoring questions correct (*Original Test*) and group tasks with all scoring questions correct (*Coherent Test*). Formally, considering that there are $N$ instructions containing Selection, they are divided into $K$ task groups. Each instruction $i$ has $m_i$ scoring questions, and the result

of the $j$-th scoring question is $r'_{ij}$ (the same definition as Section 4.2). The results of *Original Test* will be calculated as $\frac{1}{N}\sum_{i=1}^{N}(\bigwedge_{j=1}^{m_i} r'_{ij})$, and the results of *Coherent Test* will be calculated as $\frac{1}{K}\sum_{k=1}^{K}\bigwedge_{i\in Group(k)}(\bigwedge_{j=1}^{m_i} r'_{ij})$. Instructions containing Selection are categorized as either single-layer or multi-layer nested, respectively. As shown in Figure 7, for single-layer Selection instructions, LLMs with stronger instruction-following abilities show a smaller performance drop in the coherent test, which better understands the selection structure. For more complex multi-layer nested Selection instructions, even the state-of-the-art LLM, GPT-4, achieves only 14.9% accuracy in the coherent test. At the same time, smaller-scale LLMs can't perfectly follow any group of instructions. The results highlight current LLMs' weaknesses in following multi-layer tree-structured instructions.

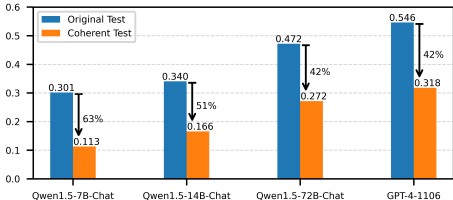 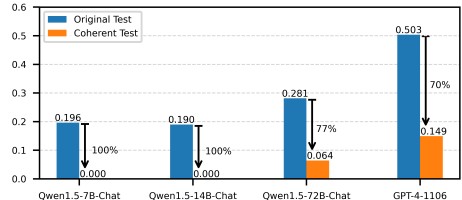

Figure 7: The performance variance under the coherent test for *Selection*. The left side represents single-layer *Selection* instructions, and the right side corresponds to multi-layer *Selection* instructions.

**Comparisons between Other Capabilities.** We compare the performance of representative LLMs across 3 prominent LLM evaluation benchmarks in addition to COMPLEXBENCH: IFEval[19], HumanEval[48], and MATH[49], focusing on instruction-following, coding, and mathematical ability, respectively. As shown in Table 7, although the performance of various LLMs on COMPLEXBENCH is well correlated with their performance on other benchmarks, the rankings of LLMs on COMPLEXBENCH do not entirely correspond with those on the other three benchmarks. For instance, ChatGLM3-6B-Chat demonstrates outstanding coding and mathematical abilities among LLMs of similar scale, but it notably struggles with complex instruction-following. On the other hand, while Llama-3-70B-Instruct surpasses GPT-4-1106 on IFEval and ranks first, it still shows a performance gap with GPT-4-1106 on COMPLEXBENCH. This discrepancy is primarily in the areas of instructions with complex constraints composition, which are not covered by IFEval, indicating that COMPLEXBENCH can provide a complementary perspective for LLM evaluation.

| Model | COMPLEXBENCH | IFEval | HumanEval | MATH |
|---|---|---|---|---|
| GPT-4-1106 | 0.800 | 75.4 | 84.6 | 64.3 |
| GLM-4 | 0.779 | 66.7 | 72.0 | 47.9 |
| Qwen1.5-72B-Chat | 0.752 | 55.8 | 71.3 | 42.5 |
| Llama-3-70B-Instruct | 0.757 | 78.9 | 81.7 | 50.4 |
| Llama-3-8B-Instruct | 0.638 | 68.6 | 62.2 | 30.0 |
| Mistral-7B-Instruct | 0.592 | 40.5 | 30.5 | 13.1 |
| Qwen1.5-7B-Chat | 0.658 | 38.8 | 46.3 | 23.2 |
| InternLM2-7B-Chat | 0.634 | 46.5 | 59.8 | 23.0 |
| ChatGLM3-6B-Chat | 0.546 | 28.1 | 64.0 | 25.7 |
| Correlation with COMPLEXBENCH | - | 0.814 | 0.715 | 0.895 |

Table 7: Model comparison on different abilities. The last row shows the Pearson correlation between the performance of LLMs in COMPLEXBENCH and other benchmarks.

## 6 Conclusion

In this work, we propose COMPLEXBENCH, a systematical benchmark for complex instruction-following. We first propose a hierarchical taxonomy for complex instructions, including 4 constraint types, 19 constraint dimensions, and 4 composition types. Furthermore, we manually collect a high-quality dataset accordingly. Along with the dataset, we propose a structure-aware automatic evaluation method for complex instruction-following with constraints composition and further enhance the evaluation accuracy by equipping LLM-based evaluators with rules. Finally, we conduct extensive experiments to evaluate the performance of current representative LLMs on complex instruction-following and uncover their significant deficiencies in dealing with complex composition types. In summary, we posit that COMPLEXBENCH can serve as a valuable tool for benchmarking the complex instruction-following ability of LLMs, providing a complementary perspective for LLM evaluation and useful insights for further work to improve this ability of LLMs.

## Acknowledgements

This work was supported by the National Science Foundation for Distinguished Young Scholars (No. 62125604), the NSFC projects (No. 62306160), and the Tsinghua University Initiative Scientific Research Program. We would also like to thank Zhipu AI for sponsoring the computation resources and annotation costs used in this work.

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

## A Limitations

The limitations of our work are summarized as follows:

**Monolingual Capability**. COMPLEXBENCH is primarily constructed based on Chinese reference instructions, which may neglect some elements in other languages and cultures that can influence the complexity of instructions. Recognizing this constraint, we plan to expand COMPLEXBENCH by incorporating multiple languages to investigate the disparities in complex instruction-following ability of LLMs across different linguistic environments in future iterations.

**LLM-based Evaluation**. The evaluation method based on LLM is widely used in the automatic evaluation process of COMPLEXBENCH. Although experiments show that our evaluation method achieves satisfactory agreement with human judgment generally, the potential biases of LLM-as-Judge, such as verbosity and self-enhancement [5], may affect the overall evaluation correctness. Additionally, we utilize GPT-4-1106 commercial APIs for evaluation, which presents challenges such as high costs and potential data leakage. We leave the development of more accurate and efficient methods for evaluating complex instruction-following as important future work.

## B Author Statement and License

COMPLEXBENCH is distributed under CC BY 4.0. The evaluation code of COMPLEXBENCH is distributed under the MIT license. We will bear all responsibility in case of violation of rights, etc.

## C Task Distribution of COMPLEXBENCH

We refer to the taxonomy of AlignBench [6] to categorize the task types of instructions in the COMPLEXBENCH. Taking into account that instructions about mathematics have relatively fixed answers and are difficult to construct complex instructions, as well as the coarse granularity of the writing ability category. We remove mathematical and use 4 subcategories of writing ability in AlignBench: practical writing, creative writing, professional writing, and custom writing. When annotators construct instructions, they also provide task category labels simultaneously, the results are shown in Table 8.

| Category | #Samples |
|---|---|
| Fundamental Language Ability | 159 |
| Advanced Chinese Understanding | 62 |
| Open-ended Questions | 115 |
| Practical Writing | 195 |
| Creative Writing | 105 |
| Professional Writing | 183 |
| Custom Writing | 73 |
| Logical Reasoning | 107 |
| Task-oriented Role Play | 95 |
| Professional Knowledge | 56 |
| Total | 1150 |

Table 8: Task distribution of COMPLEXBENCH dataset.

## D Details of Constraint Dimensions

### D.1 Lexical Constraint

1. **Word Matching.** The response should accurately find the corresponding content of certain keywords in the given instruction.

2. **Keywords.** The response should (not) include certain keywords, or include several words from a keyword list.

### D.2 Format Constraint

1. **JSON Format.** The entire response should be wrapped in JSON format.

2. **Markdown Format.** The response should follow specific Markdown formats, such as equations, headings, and tables.

3. **Bullets Format.** The response should (not) contain bullet points.

4. **Length.** Control the length of the response, including the number of words, sentences, paragraphs, etc. This constraint can be used in combination with others, such as controlling the number of bullet points or the number of keywords included.

5. **Start with.** Control the content at the beginning of the response.

6. **End with.** Control the content at the end of the response.

7. **Punctuation.** Control the punctuation that appears in the response.

8. **Template.** The response should mimic the format of the given output template.

### D.3 Semantic Constraint

1. **Language Style.** The response should adhere to a specific language style. We use the taxonomy of CharacterGLM [10], which defines language style from multiple aspects such as formality, imitation of celebrities, context-specific scenes, and discourse features (like using style from a certain website, emoji, etc.).

2. **Personalization.** The response should align with certain character attributes.

3. **Topic.** The response should focus on a specific topic.

4. **Sentiment.** The response should contain specific emotions. We refer to the six fine-grained categories of ECM [38] for sentiment, named as *Like*, *Happy*, *Sad*, *Disgust*, *Angry*, *Other*.

### D.4 Utility Constraint

1. **Helpfulness.** The response should follow task descriptions.

2. **Target Language.** The response should be in a specific language, such as simplified Chinese, traditional Chinese or English.

3. **Supportiveness.** The response should be faithful to input texts, answering based on the information provided in the text completely.

4. **Consistency.** The content of the response should be consistent and free of contradictions.

5. **Factuality.** The response should correspond with facts, which primarily applies to instructions with definitive answers such as mathematical and logical reasoning.

## E  Prompts in Rule-Augmented LLM-based Evaluation

### E.1  Prompts for Extractor in Rule-Augmented LLM-based Evaluation

Table 9 provides the prompt template we used for the LLM extractor in Rule-Augmented LLM-based evaluation. And Table 10 provides an example of scoring object extraction and their English translation. To improve performance, we use 6 manually constructed in-context examples in the prompt. Considering that the extraction of content differs significantly when there are multiple scoring objects (e.g., scoring question *"Does each shot's dialogue in the model output start with an interrogative sentence?"*), compared to when there is only one scoring object (e.g., scoring question *"Does the title of the speech given by the model have no more than 10 characters?"*). We use different sets of in-context examples for these two situations.

### E.2  Prompts for Evaluator in Rule-Augmented LLM-based Evaluation

Table 11 provides the prompt template we used for the LLM evaluator in Rule-Augmented LLM-based evaluation. And Table 12 provides an example of automatic evaluation and their English translation. We have also explored different settings where all scoring questions from the instruction are presented to the evaluation model simultaneously, or asking the evaluation model to choose "YES" or "NO" without analysis. Ultimately, we found that the current settings achieve the highest level of agreement with humans.

You are an information extraction expert. Below, you will be provided with an [Input Instruction] and its corresponding [Model Response]. Additionally, you will be given a [Scoring Question], which is designed to assess whether [Model Response] satisfies some of requirements within [Input Instruction]. Your task is to extract scoring object in [Model Response] for [Scoring Question].

For example, if [Model Response] contains two essays and [Scoring Question] is "Does the first essay have at least 500 words?", then you should only output the first essay from [Model Response]. If [Scoring Question] is "Does the second essay use a vivid language style?", then you should only output the second essay from [Model Response]. And if [Scoring Question] is "Does the output end with 'Reporter from Bloomberg'?", then you should only output the last sentence of [Model Response].

## Note
(1) You should copy continuous segments from [Model Response] exactly as it is, without any modification, addition, deletion, or splicing.
(2) Your task is not to extract the part of [Model Response] that satisfies [Scoring Question], but to extract scoring object in [Model Response] for [Scoring Question], even if it does not satisfy corresponding requirements. You do not need to pay attention to what the specific requirements of [Scoring Question] are, nor do you need to evaluate whether [Model Response] satisfies [Scoring Question] requirements.
(4) If there are multiple scoring objects in [Model Response], please use "||" to separate each other. If the scoring object is the entire [Model Response], please directly output "All". If the scoring object does not exist in [Model Response], please directly output "None".
(5) Generally, "beginning" refers to the first sentence of [Model Response], and "ending" refers to the last sentence of [Model Response].

Please first give your analysis and explanation of the task, then output the result of the evaluation object you extracted.

## Output Format
[**Explanation**]
xxx

[**Evaluation Object for Scoring Question**]
Scoring Object: xxx

{In-Context Examples}

Please refer to the above examples, and extract the scoring object in [Model Response] for [Scoring Question]. Again, note that you should copy the continuous segments from [Model Response] exactly as it is, without any modification, addition, deletion, or splicing. When giving your scoring object, you must ensure that every character is a real existence in [Model Response].

[**Input Instruction**]
{Input Instruction}

[**Model Response**]
{Model Response}

[**Scoring Question**]
{Scoring Question}

Table 9: Prompt template for extraction in Rule-Augmented LLM-based Evaluation.

| | |
|---|---|
| **Input Instruction** | 提供一个例子，说明不同职业的个人如何应用持续学习的概念来推进他们的职业生涯，要求字数不超过200字，以"例子输出完毕"结尾。然后针对这条例子生成一篇文章对其进行反驳，要求字数在500字左右，分点明确，带有激烈的情绪。 |
| | Provide an example within 200 words to illustrate how individuals from different professions can apply the concept of continuous learning to advance their careers, ending with "End of example." Then, write an article to refute this example around 500 words, clearly divided into points, and with strong emotions. |
| **Model Response** | 例子：
李华是一名软件工程师，他通过持续学习新的编程语言和技术框架来保持自己的技能更新。他利用业余时间在线学习人工智能和机器学习课程，这使他能够在工作中承担更复杂的项目，提高了他的市场竞争力。通过参加行业会议和网络研讨会，李华不仅扩大了自己的专业网络，还与行业领导者建立了联系。这些努力最终帮助他获得了一次职业晋升的机会，并在他的公司中扮演了更重要的角色。例子输出完毕。

反驳文章：
持续学习被广泛宣扬为职业发展的万能钥匙，但这种观点忽视了现实中的复杂性和个体差异。以李华的例子为起点，我们可以看到这种理想化的学习模式并不总是可行的，甚至可能是误导性的。
首先，不是每个人都有能力在紧张的工作之余抽出时间来学习新技能。李华能够在线学习新课程，这本身就是一种奢侈。对于那些需要在两份工作之间奔波，或者有家庭责任的人来说，这种持续学习的建议简直是一种讽刺。其次，学习新技能并不总能保证职业晋升或更好的工作机会。市场上充斥着高技能但失业的人才，这表明持续学习并非成功的保证。李华的例子可能是个例，而不是普遍现象。再者，过分强调个人努力的同时，忽略了职场中普遍存在的不平等和偏见。不是每个人都能通过增加技能来克服这些障碍。有时候，即使是最有才华的个体也会因为种族、性别或年龄等因素而被边缘化。最后，持续学习的压力可能导致工作与生活之间的界限模糊，引发职业倦怠和个人生活的疏远。李华可能会因为不断追求职业发展而牺牲了个人的休息时间和社交生活，这是一个不可持续的生活方式。
综上所述，持续学习并非职业成功的灵丹妙药。我们需要更加深入地考虑个人情况和社会结构，而不是简单地将个人发展的责任推给个体。在这个复杂的世界里，我们应该追求更加平衡和包容的发展路径。
Example:
Li Hua is a software engineer who keeps his skills up-to-date by continuously learning new programming languages and technology frameworks. He uses his spare time to take online courses in artificial intelligence and machine learning, which enables him to take on more complex projects at work and improves his market competitiveness. By attending industry conferences and webinars, Li Hua not only expanded his professional network but also established connections with industry leaders. These efforts ultimately helped him secure a promotion opportunity and play a more significant role in his company. End of example.

Counter-argument article:
Continuous learning is widely touted as the universal key to career development, but this perspective overlooks the complexities and individual differences in the real world. Starting with Li Hua's example, we can see that this idealized learning model is not always feasible and may even be misleading.
Firstly, not everyone can carve out time to learn new skills amidst a demanding job. The fact that Li Hua can take online courses is a luxury in itself. For those who need to juggle between two jobs or have family responsibilities, the advice of continuous learning can be nothing but satire. Secondly, learning new skills does not always guarantee career advancement or better job opportunities. The market is filled with highly skilled but unemployed talent, indicating that continuous learning is not a guarantee of success. Li Hua's example may be an exception rather than the norm. Furthermore, the emphasis on individual effort overlooks the prevalent inequality and bias in the workplace. Not everyone can overcome these obstacles by acquiring more skills. Sometimes, even the most talented individuals can be marginalized due to factors such as race, gender, or age. Lastly, the pressure to continuously learn can blur the boundaries between work and personal life, leading to burnout and alienation from one's personal life. Li Hua might sacrifice his rest and social life in the pursuit of constant career development, which is an unsustainable way of living.
In conclusion, continuous learning is not a panacea for career success. We need to delve deeper into individual circumstances and social structures, rather than simply attributing the responsibility of personal development to the individual. In this complex world, we should strive for a more balanced and inclusive path of development. |
| **Scoring Question** | 模型生成的例子是否字数不超过200字？ （输出长度）
Is the example generated by the model response within 200 words? **(Length)** |
| **Extractor Output** | 【解释】
该评分问题的评分对象是【模型回复】中的例子部分，所以需要直接输出【模型回复】中，李华作为软件工程师的例子对应部分。

【模型回复中评分问题的评测对象】
评分对象：例子：
李华是一名软件工程师，他通过持续学习新的编程语言和技术框架来保持自己的技能更新。他利用业余时间在线学习人工智能和机器学习课程，这使他能够在工作中承担更复杂的项目，提高了他的市场竞争力。通过参加行业会议和网络研讨会，李华不仅扩大了自己的专业网络，还与行业领导者建立了联系。这些努力最终帮助他获得了一次职业晋升的机会，并在他的公司中扮演了更重要的角色。例子输出完毕。
**[Explanation]**
The scoring object for this scoring question is the example section in the [Model Response], so it is necessary to directly output the corresponding part of the example with Li Hua as a software engineer in the [Model Response]

**[Evaluation Object for Scoring Question]**
Scoring Object: Example:
Li Hua is a software engineer who keeps his skills up-to-date by continuously learning new programming languages and technology frameworks. He uses his spare time to take online courses in artificial intelligence and machine learning, which enables him to take on more complex projects at work and improves his market competitiveness. By attending industry conferences and webinars, Li Hua not only expanded his professional network but also established connections with industry leaders. These efforts ultimately helped him secure a promotion opportunity and play a more significant role in his company. End of example. |
| **Rule** | *len* ({Evaluation Object for Scoring Question}) < 200 ? |
| **Evaluation Result** | 1 (Yes) |

Table 10: An example of segments extraction and their English translation.

Please act as a fair judge, analyze the content of the Model Response, and choose "YES" or "NO" to answer whether the requirement of the Question is satisfied. You should follow the following judgment rules.

- The Question can be seen as the scoring points of the Instruction in steps, judging whether a part of it is satisfied. Therefore, you only need to consider the requirement within the Question, without focusing on whether the entire Instruction is fully satisfied.
- YES: Check whether the Model Response completes the requirement of the Question thoroughly. You should fully understand the meaning of the Question and not miss any small details, only focus on the Question and not pay attention to other requirements in the Instruction. It must be perfectly and sufficiently completed to be evaluated as "YES", without any slight errors or ambiguities. There should not be situations such as "basically correct", "mostly correct", or "correct under certain conditions". These situations should all be evaluated as "NO".

- NO: If the Model Response does not satisfy the requirement of the Question or provides relevant information about the Question, choose "NO".
**Example**: If the Question asks "Is the second sentence of the generated text a complex sentence?" but the Model Response only has one sentence. It does not provide relevant information about the Question. Therefore, you should choose "NO".

## Detailed Scoring Rules
(1) When you evaluate whether the Model Response contains bullet points, it must have clear bullet points or numbers to be evaluated as "YES". Merely using conjunctions like "firstly", "then", "next", and "finally" cannot be considered bullet points, and should be evaluated as "NO".
(2) When you evaluate whether the Model Response is in a specific language (such as Chinese/English) unless the Instruction mentions the need to use multiple languages, it must use only that language to be evaluated as "YES", the appearance of other languages (i.e., words from other languages) should be evaluated as "NO".
(3) When you evaluate whether the Model Response selects the correct branch, it is necessary to judge whether the Model Response completes the sub-task of the corresponding branch based on the selection branch in the Instruction.
(4) If the Question includes descriptions like "every", "all", etc., you should consider every object in Model Response about the Question, only if all objects satisfy the requirement of the Question, it can be evaluated as "YES".

## Output Format
Analysis: xxx
Answer: Yes / No

## Evaluation Information
**Instruction**
{Input Instruction}

**Model Response**
{Model Response}

**Question**
{Scoring Question}

Please analyze and answer whether the Model Response satisfies the requirement of Question:

Table 11: Prompt template for LLM-based evaluation in Rule-Augmented LLM-based Evaluation.

| | |
|---|---|
| **Input Instruction** | 写一篇关于美国政治新闻的博客文章，标题用双角括号标示，即«标题»，并且确保句子数量少于5句（不包括5句）。句子应当较长，以使你的回答总字数达到或超过250字。

Write a blog post about American political news with the title indicated by double angle brackets, i.e. «Title», and ensure that the number of sentences is less than 5. The sentences should be relatively long to make the total number of words in your response reach 250. |
| **Model Response** | «美国政治新闻：分裂与变革的交响曲»

在2022年的美国，政治新闻如同一部紧张的交响曲，旋律中交织着分裂与变革的强烈节奏。民主党和共和党的分歧日益加剧，疫苗接种的鸿沟、经济复苏的不平等以及社会正义的呼唤都在琴弦上弹出激昂的和弦。川普总统的退场并未平息争议，而是开启了拜登政府的新篇章，他的"重建美国"计划在国会中遭遇了重大挑战，立法进程中的僵局成为新闻焦点。与此同时，地方选举中的基层力量崛起，显示选民对传统两党制的不满正在发酵。这场政治交响乐，每个音符都预示着未来走向的不确定性，但无论如何，美国政治的戏剧性永远不会缺少观众。

«American Political News: Symphony of Division and Transformation»

In 2022, American political news is like a tense symphony, with a melody interwoven with the strong rhythms of division and change. The divide between the Democratic and Republican parties is intensifying, and the chasms in vaccine distribution, unequal economic recovery, and calls for social justice all play passionate chords on the strings. The departure of President Trump has not quelled the controversy but has opened a new chapter for the Biden administration. His "Build Back Better" plan has faced significant challenges in Congress, with the deadlock in the legislative process becoming a focus of the news. Meanwhile, grassroots forces are rising in local elections, showing that voters' dissatisfaction with the traditional two-party system is fermenting. Every note in this political symphony foretells the uncertainty of the future, but no matter what, the drama of American politics will never lack an audience. |
| **Scoring Question** | 输出内容是否是关于美国政治新闻的博客文章？（满足用户需求）
Is the model response a blog post about American political news? **(Helpfulness)** |
| **Evaluator Output** | 分析：根据模型回复的内容，可以看到它确实是一篇关于美国政治新闻的博客文章。开头使用了双角括号标示的标题《美国政治新闻：分裂与变革的交响曲》，接着以较长的句子描述了美国政治的现状，提及了与政治相关的关键词汇，如民主党、共和党、川普总统、拜登政府、立法进程、地方选举等。因此，模型回复确实是一篇关于美国政治新闻的博客文章。
答案：是

Analysis: Based on the content of the [Model Response], it can be seen that it is indeed a blog post about American political news. The response starts with a title enclosed in double angle brackets, "American Political News: Symphony of Division and Transformation," followed by longer sentences describing the current state of American politics, mentioning keywords related to politics such as Democrats, Republicans, President Trump, the Biden administration, legislative processes, local elections, etc. Therefore, the [Model Response] is indeed a blog post about American political news.
Answer: Yes |
| **Evaluation Result** | 1 (Yes) |

Table 12: An example of automatic evaluation with evaluator LLM and their English translation.

# F Detailed Information about Human Annotation

We recruited 12 college students for data annotation of ComplexBench. As for annotators training, all the annotators are required to complete a training tutorial that includes 50 samples. We provide feedback to help them calibrate the annotation criteria. In addition, during the annotation process, we conduct spot checks on annotated data. We also retrain the annotators based on the issues found in their annotations and require them to recheck such issues in their annotated data. The annotator training process for dataset creation and human evaluation is similar. And the total labor cost is approximately 2800$. We provide the guidelines for each annotation task as follows.

## F.1 Guidelines for Data Annotation

This section corresponds to Section 4.1 (**Data Annotation and Validation**), where the guidelines for data annotation are shown in Table 13, and Table 14 presents the English translation version. In this annotation task, annotators will be provided with reference instructions, the requirements of the minimum number of constraint dimensions in each constraint type, and the minimum number of composition types. Annotators are instructed to construct new complex instructions based on the reference instructions while annotating all the constraint dimensions and composition types within the newly constructed instructions. Then, they are also required to annotate scoring questions for the newly constructed instructions, and the task type of the newly constructed instructions.

## F.2 Guidelines for Selection Branch Expansion

This section corresponds to Section 4.1 (**Selection Branch Expansion**), where the guidelines for selection branch expansion are shown in Table 15, and Table 16 presents the English translation version. In this annotation task, instructions with *Selection* that are annotated in the above task will be provided to the annotators. Annotators are instructed to modify the selection conditions of the original instructions and construct several new instructions to cover all the different selection branches apart from the original instructions. For each new instruction, all the information required by the above annotation task needs to be annotated.

## F.3 Guidelines for Overall Preference Annotation

This section corresponds to Section 5.1 (**Agreement Evaluation**), where the guidelines for overall preference annotation and their English translation are shown in Table 17. Given an instruction and two model responses (denoted as A and B), the human annotators are instructed to compare the quality and choose from 3 options, namely A better than B, tie, and B better than A.

## F.4 Guidelines for Scoring Questions Verification

This section corresponds to Section 5.1 (**Agreement Evaluation**), where the guidelines for scoring questions verification and their English translation are shown in Table 18. Given an instruction and a corresponding model response, as well as a scoring question for the instruction, human annotators are instructed to judge whether the requirements of the scoring question are satisfied by the model response.

## F.5 Guidelines for Instruction Decomposition

This section corresponds to Section 5.2.3 (**Decomposition of instructions with composition types**), where the guidelines for instruction decomposition and their English translation are shown in Table 19. Given an instruction containing composition types, human annotators are instructed to decompose the instruction based on composition types (e.g., *Chain* into sequential tasks, *Selection* into selection and execution branches, while *And* remains intact) and split the scoring questions of original instructions into corresponding decomposed instructions.

# G An Example of Data Construction

We provide a specific example (translated into English) about data annotation in Table 20 to facilitate a better understanding of the process. The objectives of the first two steps in data construction, i.e.,

**Reference Instruction Collection** and **Task Allocation** (as mentioned in Section 4.1) are to obtain **Reference Instruction** and **Task Requirements** fields of the table, respectively. Each field of the table corresponds to Table 13.

---

下面将给你提供一条参考指令，四个约束类型各自的约束维度数量要求，以及组合方式数量要求，请你依次完成如下标注任务。

1. 基于参考指令构造出一条新的复杂指令，要求该指令中包含的各约束类型的约束维度数量大于或等于要求数量，该指令中包含的各组合方式的数量大于或等于要求数量。你也可以不参考参考指令从头开始编写。
2. 标注新构造的复杂指令的任务类型，在十个类型中选择最接近的一类。
3. 标注新构造的复杂指令中包含的所有约束维度，组合方式。
4. 标注新构造的复杂指令的得分问题。请针对该指令中的每一个约束维度、组合方式，分别设计一个可以用"是/否"回答的得分问题判定其是否得到满足。
5. 标注得分问题之间的依赖关系。我们规定，对于链式组合方式，后续任务的所有得分问题依赖于判断前序任务是否完成的得分问题；对于选择组合方式，判断正确分支执行情况的所有得分问题依赖于判断正确分支是否被选择的得分问题。在标注每个得分问题时，同时需要标注其依赖于哪些得分问题。

在构造复杂指令时，必须遵循如下三个总体原则：
1. 合理性：指令必须没有歧义，有相对明确的正确答案。
2. 约束有效性：指令中包含的每一个约束维度，均应该对输出产生实质影响。
3. 难度：指令必须是足够困难的，原则上应该比参考指令更为复杂。

[参考指令]
{reference_instruction}

[任务要求]
词级约束最少数量：{number_of_lexical_constraints}
格式约束最少数量：{number_of_format_constraints}
语义约束最少数量：{number_of_semantic_constraints}
整体约束最少数量：{number_of_utility_constraints}

并列组合最少数量：{number_of_And}
链式组合最少数量：{number_of_Chain}
选择组合最少数量：{number_of_Selection}

请你根据参考指令构造出新的复杂指令：{newly_constructed_instruction}
请选择构造指令的任务分类，从以下十项中选择一项：{task_type_of_newly_constructed_instruction}
A. 基本能力 B. 中文理解 C. 综合问答 D. 实用文本写作 E. 创意写作 F. 专业文本写作 G. 个性化写作 H.逻辑推理
I.角色扮演 J. 专业能力

请选择构造指令中包含的所有词级约束，多选，一个选项可以选择多次：{lexical_constraints_in_newly_constructed_instruction}
A. 输入词匹配 B. 输出关键词

请选择构造指令中包含的所有格式约束，多选，一个选项可以选择多次：{format_constraints_in_newly_constructed_instruction}
A. Json格式 B. Markdown格式 C. 分点格式 D. 标点格式 E. 输出长度 F. 开头格式 G. 结尾格式 H. 基于模板格式

请选择构造指令中包含的所有语义约束，多选，一个选项可以选择多次：{semantic_constraints_in_newly_constructed_instruction}
A. 语言风格 B. 角色属性 C. 话题 D. 情感

请选择构造指令中包含的所有整体约束，多选，一个选项可以选择多次：{utility_constraints_in_newly_constructed_instruction}
A. 目标语言 B. 支持性 C. 连贯性 D. 事实正确性 E. 满足用户需求

请选择构造指令中包含的所有组合方式，多选，一个选项可以选择多次：{composition_types_in_newly_constructed_instruction}
A. 并列 B. 链式 C. 选择

请标注构造指令的所有评分问题，同时标注出其评测的约束维度/组合方式，每个得分问题参考如下例子编写：
**1**. 模型生成的文章语言是否为英文？（目标语言）
{scoring_questions_for_newly_constructed_instruction}

请标注评分问题之间的依赖关系，每个得分问题参考如下例子编写（没有依赖关系则不必编写）：
**4**. 模型回复字数是否在300字左右？（输出长度，依赖于**1**）
{dependencies_of_scoring_questions}

---

Table 13: Guidelines for data annotation. The blue part is the information provided to the annotators, and the red part is content that requires the annotators to make annotations.

# H English Translation of COMPLEXBENCH

Since the data we collected from real-world application scenarios is primarily in Chinese, we initially use our data construction pipeline to construct Chinese data. Subsequently, we use GLM-4[41] to translate the constructed data into English and manually correct any errors in the translation to produce the English version of COMPLEXBENCH.

Below, a reference instruction, the requirements of the minimum number of constraint dimensions in each constraint type, and the minimum number of composition types will be provided. Please complete the following annotation tasks in order.

1. Construct a new complex instruction based on the reference instruction, ensuring that the number of constraint dimensions in each constraint type within the instruction is greater than or equal to the requirements, and the number of composition types within the instruction is greater than or equal to the requirements. You may also create the new complex instruction from scratch without referencing the provided reference instruction.
2. Annotate the task type of the newly constructed complex instruction, choosing the closest type from the ten options provided.
3. Annotate all the constraint dimensions and composition types within the newly constructed complex instruction.
4. Annotate the scoring questions for the newly constructed complex instruction. Please design a "yes/no" question for each constraint dimension and composition type to verify if it is satisfied.
5. Annotate the dependencies of scoring questions. Specifically, for *Chain*, all the scoring questions of the subsequent task depend on the answers to those of the preceding task. And for *Selection*, all the scoring questions of the selection branch depend on whether the correct selection branch is selected. When annotating each scoring question, please also annotate which scoring questions it depends on (if any).

When constructing complex instructions, you should adhere to the following three general principles:
1. Clarity & Reasonableness: The instruction should be easy to understand, unambiguous, and realistic, with at least one reasonable answer.
2. Validity of Constraints: Every constraint within the instruction should substantially influence the output.
3. Complexity & Difficulty: The instruction should be challenging for most LLMs and be capable of distinguishing the complex instruction-following abilities of different LLMs.

**[Reference Instruction]**
{reference_instruction}

**[Task Requirements]**
The minimum number of lexical constraints: {number_of_lexical_constraints}
The minimum number of format constraints: {number_of_format_constraints}
The minimum number of semantic constraints: {number_of_semantic_constraints}
The minimum number of utility constraints: {number_of_utility_constraints}

The minimum number of *And*: {number_of_And}
The minimum number of *Chain*: {number_of_Chain}
The minimum number of *Selection*: {number_of_Selection}

Please construct a new complex instruction based on the reference instruction: {newly_constructed_instruction}
Please choose the task category for the constructed instruction from the following ten options: {task_type_of_newly_constructed_instruction}
A. Fundamental Language Ability    B. Advanced Chinese Understanding    C. Open-ended Questions    D. Practical Writing    E. Creative Writing    F. Professional Writing    G. Custom Writing    H. Logical Reasoning    I. Task-oriented Role Play    J. Professional Knowledge

Please choose all lexical constraints within the constructed instruction, multiple selections are allowed, and an option can be chosen more than once: {lexical_constraints_in_newly_constructed_instruction}
A. Word Matching    B. Keywords

Please choose all format constraints within the constructed instruction, multiple selections are allowed, and an option can be chosen more than once: {format_constraints_in_newly_constructed_instruction}
A. Json Format    B. Markdown Format    C. Bullets Format    D. Punctuation    E. Length    F. Start with    G. End with    H. Template

Please choose all semantic constraints within the constructed instruction, multiple selections are allowed, and an option can be chosen more than once: {semantic_constraints_in_newly_constructed_instruction}
A. Language Style    B. Personalization    C. Topic    D. Sentiment

Please choose all utility constraints within the constructed instruction, multiple selections are allowed, and an option can be chosen more than once: {utility_constraints_in_newly_constructed_instruction}
A. Target Language    B. Supportiveness    C. Consistency    D. Factuality    E. Helpfulness

Please choose all composition types within the constructed instruction, multiple selections are allowed, and an option can be chosen more than once: {composition_types_in_newly_constructed_instruction}
A. *And*    B. *Chain*    C. *Selection*

Please annotate all scoring questions for the constructed instruction, and indicate the constraint dimensions/composition types they evaluate. Each scoring question should be formatted as follows:
**1**. Is the language of the article generated by the model in English? (Target language)
{scoring_questions_for_newly_constructed_instruction}

Please annotate the dependencies of scoring questions. Each scoring question should be formatted as follows (no need to write if there is no dependency):
**4**. Is the number of words in the model's response more than 300? (Length, depends on **1**)
{dependencies_of_scoring_questions}

Table 14: Guidelines for data annotation (translated into English). The blue part is the information provided to the annotators, and the red part is content that requires the annotators to make annotations.

下面将给你提供一条包含选择逻辑的指令，请你保持该指令中所有的选择分支不变，仅修改该指令中选择函数的选择条件，构造若干条新指令，覆盖与原指令不同的所有选择分支。例如，对于单层的选择逻辑，选择函数有**M**个不同的取值，则你应该构造**M-1**条新指令，改变选择条件以覆盖选择函数与原指令不同的所有取值。

在构造出新指令后，和数据构造任务相同，你需要标注出新构造指令中的任务类型，包含的所有约束维度、组合方式，并编写新构造指令的评分问题和其互相之间的依赖关系。

[原指令]
{instruction}

请你填写根据原指令，修改选择条件构造的第一条新指令：{newly_constructed_instruction_1}
请选择构造指令的任务分类，从以下十项中选择一项：{task_type_of_newly_constructed_instruction_1}
A. 基本能力  B. 中文理解  C. 综合问答  D. 实用文本写作  E. 创意写作  F. 专业文本写作  G. 个性化写作  H.逻辑推理
I.角色扮演  J. 专业能力

请选择构造指令中包含的所有词级约束，多选，一个选项可以选择多次：{lexical_constraints_in_newly_constructed_instruction_1}
A. 输入词匹配  B. 输出关键词

请选择构造指令中包含的所有格式约束，多选，一个选项可以选择多次：{format_constraints_in_newly_constructed_instruction_1}
A. Json格式  B. Markdown格式  C. 分点格式  D. 标点格式  E. 输出长度  F. 开头格式  G. 结尾格式  H. 基于模板格式

请选择构造指令中包含的所有语义约束，多选，一个选项可以选择多次：{semantic_constraints_in_newly_constructed_instruction_1}
A. 语言风格  B. 角色属性  C. 话题  D. 情感

请选择构造指令中包含的所有整体约束，多选，一个选项可以选择多次：{utility_constraints_in_newly_constructed_instruction_1}
A. 目标语言  B. 支持性  C. 连贯性  D. 事实正确性  E. 满足用户需求

请选择构造指令中包含的所有组合方式，多选，一个选项可以选择多次：{composition_types_in_newly_constructed_instruction_1}
A. 并列  B. 链式  C. 选择

请标注构造指令的所有评分问题，同时标注出其评测的约束维度/组合方式，每个得分问题参考如下例子编写：
**1**. 模型生成的文章语言是否为英文？（目标语言）
{scoring_questions_for_newly_constructed_instruction_1}

请标注评分问题之间的依赖关系，每个得分问题参考如下例子编写（没有依赖关系则不必编写）：
**4**. 模型回复字数是否在300字左右？（输出长度，依赖于**1**）
{dependencies_of_scoring_questions}

……

与以上格式相同，请你填写根据原指令，修改选择条件构造的第n条新指令：{newly_constructed_instruction_n}
……

Table 15: Guidelines for selection branch expansion. The blue part is the information provided to the annotators, and the red part is content that requires the annotators to make annotations.

Below, an instruction containing *Selection* will be provided. Please keep all selection branches unchanged, and only modify the selection condition based on the selection function to construct multiple new instructions, covering all selection branches different from the original instruction. For example, for single-layer *Selection*, if the selection function has M different values, you should construct M-1 new instructions, changing the selection conditions to cover all values different from the original instruction.

After constructing the new instructions, similar to the data annotation task, you need to annotate the task types of the instructions and all constraint dimensions and composition types within the instructions. Furthermore, you should annotate the scoring questions for the newly constructed instructions and their dependencies.

**[Original Instruction]**
{instruction}

Please annotate the first new instruction by modifying the selection conditions of the original instruction: {newly_constructed_instruction_1}
Please choose the task category for the constructed instruction from the following ten options: {task_type_of_newly_constructed_instruction_1}
A. Fundamental Language Ability    B. Advanced Chinese Understanding    C. Open-ended Questions    D. Practical Writing    E. Creative Writing    F. Professional Writing    G. Custom Writing    H. Logical Reasoning    I. Task-oriented Role Play    J. Professional Knowledge

Please choose all lexical constraints within the constructed instruction, multiple selections are allowed, and an option can be chosen more than once: {lexical_constraints_in_newly_constructed_instruction_1}
A. Word Matching    B. Keywords

Please choose all format constraints within the constructed instruction, multiple selections are allowed, and an option can be chosen more than once: {format_constraints_in_newly_constructed_instruction_1}
A. Json Format    B. Markdown Format    C. Bullets Format    D. Punctuation    E. Length    F. Start with    G. End with    H. Template

Please choose all semantic constraints within the constructed instruction, multiple selections are allowed, and an option can be chosen more than once: {semantic_constraints_in_newly_constructed_instruction_1}
A. Language Style    B. Personalization    C. Topic    D. Sentiment

Please choose all utility constraints within the constructed instruction, multiple selections are allowed, and an option can be chosen more than once: {utility_constraints_in_newly_constructed_instruction_1}
A. Target Language    B. Supportiveness    C. Consistency    D. Factuality    E. Helpfulness

Please choose all composition types within the constructed instruction, multiple selections are allowed, and an option can be chosen more than once: {composition_types_in_newly_constructed_instruction_1}
A. *And*    B. *Chain*    C. *Selection*

Please annotate all scoring questions for the constructed instruction, and indicate the constraint dimensions/composition types they evaluate. Each scoring question should be formatted as follows:
**1**. Is the language of the article generated by the model in English? (Target language)
{scoring_questions_for_newly_constructed_instruction_1}

Please annotate the dependencies of scoring questions. Each scoring question should be formatted as follows (no need to write if there is no dependency):
**4**. Is the number of words in the model's response more than 300? (Length, depends on **1**)
{dependencies_of_scoring_questions}

......

In the same format as above, please annotate the nth new instruction constructed by modifying the selection conditions of the original instruction: {newly_constructed_instruction_n}
......

Table 16: Guidelines for selection branch expansion (translated into English). The blue part is the information provided to the annotators, and the red part is content that requires the annotators to make annotations.

下面将给你提供一条指令和对应的两个模型回复a、b，请你判断模型回复a、b哪个更好地遵循了指令的要求，质量更高，给出win，lose，tie的标注（win表示回复a更好）。

任务细则
1. 如果两个回复均质量很低，如完全没有理解指令要求，答非所问等情况，可以标注为tie，不需要进行过于细致地区分，但并非仅在此情况下才可以标注tie。
2. 不应该过于关注回复长度，遵循指令要求更为重要。

[指令]
{instruction}

[回复 A]
{response_a}

[回复 B]
{response_b}

你的选择是：{option}
A. win
B. tie
C. lose
简单叙述选择的理由：{explanation}

---

Below, an instruction and two corresponding model responses, a and b, will be provided. Please judge which model response better follows the instruction's requirements and is of higher quality, and choose 'win', 'lose', or 'tie' ('win' indicates response a is better).

**Task Details**
1. If both responses are of low quality, such as completely misunderstanding the instruction's requirements or being irrelevant, you can choose 'tie' and there is no need for detailed distinction, but 'tie' is not limited to this situation only.
2. Do not overly focus on the length of the responses. Following the requirements of instruction is more important.

**[Instruction]**
{instruction}

**[Response A]**
{response_a}

**[Response B]**
{response_b}

Your choice is: {option}
A. win
B. tie
C. lose
Briefly state the reason for your choice: {explanation}

Table 17: Guidelines for overall preference annotation and their English translation. The blue part is the information provided to the annotators, and the red part is content that requires the annotators to make annotations.

下面将给你提供一条指令，对应的一个模型回复，以及该指令的一个得分问题，你的任务是判断模型回复是否满足得分问题要求，标注"是"或者"否"。

任务细则
1. "是"的定义必须是完全充分地完成了得分点要求，任何存在错误、不明确、无法判断地回答都应该判定为"否"。不存在"基本上正确"，"部分条件下正确"的说法，这些情况均标注为"否"。如果模型回复中不存在得分问题所评判的对象，也应该标注为"否"。
2. 只需要考虑该得分点是否完全被模型回复满足，而不需要考虑整个输入指令是否被模型输出满足。

{in-context examples}

[指令]
{instruction}

[模型回复]
{model_response}

[得分问题]
{scoring_question}

你的选择是：{option}
A. 是
B. 否

---

Below, an instruction, a corresponding model response, and a scoring question for the instruction will be provided. Your task is to verify whether the model response meets the requirements of the scoring question, and choose "Yes" or "No."

**Task Details**
1. A "Yes" must indicate that the scoring point has been fully and sufficiently satisfied. Any response that contains errors, ambiguity, or cannot be judged should be labeled as "No". There is no such thing as "basically correct" or "correct under certain conditions". These should all be labeled as "No". If the model response does not contain the object to be evaluated by the scoring question, it should also be labeled as "No".
2. Please only consider whether the scoring point has been fully satisfied by the model response, without the need to consider whether the entire instruction has been satisfied by the model response.

{in-context examples}

**[Instruction]**
{instruction}

**[Model Response]**
{model_response}

**[Scoring Question]**
{scoring_question}

Your choice is: {option}
A. Yes
B. No

Table 18: Guidelines for scoring questions verification and their English translation. The blue part is the information provided to the annotators, and the red part is content that requires the annotators to make annotations.

下面将给你提供一条指令和该指令对应的得分问题，请你根据该指令中包含的组合方式，将该指令拆分为多条原子指令，要求每条原子指令不含有除了并列之外的其他组合方式。

请将原指令拆分为多条多轮追问形式的原子指令。对于包含链式组合方式的指令，请按每个子任务分别拆分，如果每个子任务中仍然包含组合方式，需要继续进行拆分；对于包含选择组合方式的指令，拆分后其中一条指令为选择正确分支，另一部分指令为执行正确分支，正确分支中仍包含组合方式的，需要继续进行拆分。如果该指令难以拆分的，可以选择跳过该指令不进行拆分。

在你完成指令拆分后，还需要将原指令的得分问题分配到对应的原子指令中。禁止增添或修改得分问题，仅允许在不改变原意的情况下，为增加流畅性对得分问题进行微量修改。

{in-context examples}

[原指令]
{instruction}

[得分问题]
{scoring_questions}

请填写你拆分原指令得到的第一条指令：{sub_instructions_0}
请填写该指令对应的得分问题，必须选取原指令得分问题中对应的连续片段：{scoring_questions_for_sub_instructions_0}

请填写你拆分原指令得到的第二条指令（没有则不必填写）：{sub_instructions_1}
请填写该指令对应的得分问题，必须选取原指令得分问题中对应的连续片段：{scoring_questions_for_sub_instructions_1}

……

请填写你拆分原指令得到的第n条指令（没有则不必填写）：{sub_instructions_n}
请填写该指令对应的得分问题，必须选取原指令得分问题中对应的连续片段：{scoring_questions_for_sub_instructions_n}

Below is an instruction and its corresponding scoring questions. Please decompose the instruction into multiple atomic instructions according to the composition types it contains, ensuring that each atomic instruction does not contain any composition types other than *And*.

Please decompose the original instruction into multiple atomic instructions in the form of a multi-turn interactive. For instructions containing *Chain*, decompose them by each task separately. If each task still contains composition types other than *And*, continue to decompose further. For instructions containing *Selection*, one of the decomposed instructions should be the choice of the correct branch, and the other part should be the execution of the correct branch. If the correct branch still contains composition types other than *And*, continue to decompose further. If an instruction is difficult to decompose, you may choose to skip and not decompose it.

After you have completed the decomposition of the instructions, you need to assign the scoring questions of the original instruction to the corresponding atomic instructions. Adding or deleting scoring questions is prohibited. Only minimal modifications to increase fluency without changing the original meaning are allowed.

{in-context examples}

**[Original Instruction]**
{instruction}

**[Scoring Questions]**
{scoring_questions}

Please annotate the first atomic instruction obtained by decomposing the original instruction: {sub_instructions_0}
Please annotate the corresponding scoring question for this instruction, making sure to select the continuous segment from scoring questions of the original instruction: {scoring_questions_for_sub_instructions_0}

Please annotate the second atomic instruction obtained by decomposing the original instruction (if any): {sub_instructions_1}
Please annotate the corresponding scoring question for this instruction, making sure to select the continuous segment from scoring questions of the original instruction: {scoring_questions_for_sub_instructions_1}

……

Please annotate the nth atomic instruction obtained by decomposing the original instruction (if any): {sub_instructions_n}
Please annotate the corresponding scoring question for this instruction, making sure to select the continuous segment from scoring questions of the original instruction: {scoring_questions_for_sub_instructions_n}

Table 19: Guidelines for instruction decomposition and their English translation. The blue part is the information provided to the annotators, and the red part is content that requires the annotators to make annotations.

| | The Information Provided to the Annotators |
|---|---|
| **Reference Instruction** | Write lyrics for a song about innovation that appeals to teenagers. Please place your entire song's lyrics in double quotation marks. |
| **Task Requirements** | The minimum number of lexical constraints: 0
The minimum number of format constraints: 2
The minimum number of semantic constraints: 1
The minimum number of utility constraints: 0

The minimum number of *And*: 0
The minimum number of *Chain*: 1
The minimum number of *Selection*: 0 |
| | The Content that Requires the Annotators to Make Annotations |
| **Newly Constructed Instruction** | Write lyrics for a song with the topic of "Beijing tourism." Please place your entire song's lyrics in double quotation marks. Then write two comments for the song, strongly conveying appreciation for it, each comment should be around 20 words. |
| **Task Type** | Creative Writing |
| **Constraint Dimensions** | Punctuation, Length, Topic, Sentiment, Helpfulness |
| **Composition Type** | *And*, *Chain* |
| **Scoring Questions and Their Dependencies** | 1. Does the model generate lyrics for a song? (*Chain*, Helpfulness)
2. Do the lyrics for the song generated by the model with the topic of "Beijing tourism"? (Topic)
3. Are all the lyrics of the song generated by the model placed in double quotation marks? (Punctuation)
4. After generating the lyrics of the song, does the model generate two comments for the song? (Helpfulness, dependent on 1)
5. Do the comments generated by the model strongly convey appreciation for the song? (Sentiment, dependent on 1)
6. Are the comments generated by the model around 20 words each? (Length, dependent on 1) |

Table 20: An example of data annotation.

# I  More Experimental Results and Analysis

## I.1  The Influence of Composition Types Nested Methods

Table 21 presents DRFR of GPT-3.5-Turbo-1106 on instructions with different numbers of each composition type. Nested multiple *Selection* seems to be significantly more difficult than other composition type nested methods. And the addition of *And* has a limited impact on the overall complexity of instructions. The result reveals the weakness in the ability of LLMs to follow complex instructions with multi-layer tree structures, highlighting the importance of further efforts to improve LLMs in these areas.

| | Composition Type | | | DRFR |
|---|---|---|---|---|
| Number | And | Chain | Selection | |
| 1 | 1 | 0 | 0 | 0.845 |
| | 0 | 1 | 0 | 0.686 |
| | 0 | 0 | 1 | **0.682** |
| 2 | 1 | 1 | 0 | 0.630 |
| | 1 | 0 | 1 | 0.651 |
| | 0 | 1 | 1 | 0.570 |
| | 0 | 0 | 2 | **0.377** |
| 3 | 1 | 1 | 1 | 0.529 |
| | 1 | 0 | 2 | 0.515 |
| | 0 | 1 | 2 | **0.308** |
| 4 | 1 | 1 | 2 | 0.083 |

Table 21: DRFR of GPT3.5-Turbo-1106 on instructions with different numbers of each composition type.

## I.2  Detailed Results of Each Constraint and Composition Type

Table 22 presents the average accuracy of LLMs on diverse constraint dimensions and composition types. Topic, Markdown Format, Consistency, Sentiment, and Personalization seem to be the easiest constraint dimensions for LLMs overall, while Length, Punctuation, Keywords, End with, and Factuality pose the greatest challenges. It is worth noting that the performance of all LLMs on Length is far from satisfactory, with even the strongest model achieving only an accuracy rate of 0.532. This result indicates that there is still significant room for improvement in the ability of current LLMs to precisely control and plan the output content.

Large Language Models: (M0) GPT-4-1106 (M1) Claude-3-Opus (M2) GLM-4 (M3) ERNIEBot-4 (M4) GPT-3.5-Turbo-1106 (M5) Qwen1.5-72B-Chat (M6) Llama-3-70B-Instruct (M7) InternLM2-20B-Chat (M8) Qwen1.5-14B-Chat (M9) Baichuan2-13B-Chat (M10) Llama-3-8B-Instruct (M11) Mistral-7B-Instruct (M12) Qwen1.5-7B-Chat (M13) InternLM2-7B-Chat (M14) ChatGLM3-6B-Chat

| | M0 | M1 | M2 | M3 | M4 | M5 | M6 | M7 | M8 | M9 | M10 | M11 | M12 | M13 | M14 | Avg. |
|---|---|---|---|---|---|---|---|---|---|---|---|---|---|---|---|---|
| **Lexical Constraint** | | | | | | | | | | | | | | | | |
| Word Matching | **0.856** | 0.847 | 0.829 | 0.757 | 0.658 | 0.811 | 0.775 | 0.793 | 0.631 | 0.649 | 0.658 | 0.622 | 0.658 | 0.712 | 0.604 | 0.729 |
| Keywords | **0.738** | 0.690 | 0.718 | 0.718 | 0.634 | 0.699 | 0.614 | 0.625 | 0.583 | 0.496 | 0.423 | 0.451 | 0.549 | 0.561 | 0.485 | 0.606 |
| Avg. | **0.766** | 0.727 | 0.745 | 0.727 | 0.639 | 0.725 | 0.652 | 0.665 | 0.594 | 0.532 | 0.479 | 0.491 | 0.575 | 0.597 | 0.513 | 0.635 |
| **Format Constraint** | | | | | | | | | | | | | | | | |
| Json Format | **0.978** | 0.822 | 0.756 | 0.800 | 0.889 | 0.778 | 0.778 | 0.778 | 0.689 | 0.689 | 0.778 | 0.711 | 0.756 | 0.667 | 0.644 | 0.779 |
| Markdown Format | 0.943 | 0.925 | **0.962** | 0.906 | 0.906 | 0.925 | 0.868 | 0.849 | 0.811 | 0.642 | 0.792 | 0.830 | 0.830 | 0.868 | 0.660 | 0.856 |
| Bullets Format | 0.828 | 0.859 | **0.865** | 0.761 | 0.779 | 0.779 | 0.828 | 0.736 | 0.663 | 0.601 | 0.650 | 0.583 | 0.638 | 0.718 | 0.558 | 0.729 |
| Punctuation | 0.738 | **0.862** | 0.569 | 0.492 | 0.631 | 0.615 | 0.662 | 0.431 | 0.538 | 0.508 | 0.646 | 0.446 | 0.477 | 0.508 | 0.354 | 0.576 |
| Length | 0.438 | 0.455 | 0.490 | **0.532** | 0.433 | 0.446 | 0.394 | 0.354 | 0.421 | 0.332 | 0.332 | 0.329 | 0.406 | 0.359 | 0.342 | 0.409 |
| Start with | 0.806 | **0.819** | 0.764 | 0.764 | 0.722 | 0.750 | 0.681 | 0.694 | 0.597 | 0.639 | 0.667 | 0.500 | 0.625 | 0.625 | 0.583 | 0.691 |
| End with | 0.766 | **0.781** | 0.703 | 0.672 | 0.750 | 0.672 | 0.734 | 0.563 | 0.531 | 0.469 | 0.656 | 0.609 | 0.531 | 0.484 | 0.469 | 0.634 |
| Template | **0.875** | 0.830 | 0.761 | 0.784 | 0.716 | 0.716 | 0.705 | 0.716 | 0.693 | 0.580 | 0.568 | 0.545 | 0.636 | 0.591 | 0.466 | 0.688 |
| Avg. | 0.669 | **0.679** | 0.658 | 0.652 | 0.623 | 0.619 | 0.604 | 0.545 | 0.550 | 0.479 | 0.523 | 0.478 | 0.537 | 0.523 | 0.450 | 0.579 |
| **Semantic Constraint** | | | | | | | | | | | | | | | | |
| Language Style | 0.812 | 0.828 | **0.834** | 0.777 | 0.694 | 0.818 | 0.787 | 0.666 | 0.768 | 0.608 | 0.691 | 0.653 | 0.758 | 0.570 | 0.513 | 0.725 |
| Personalization | 0.850 | 0.850 | 0.858 | 0.827 | 0.756 | 0.850 | **0.866** | 0.772 | 0.819 | 0.717 | 0.756 | 0.748 | 0.827 | 0.772 | 0.598 | 0.791 |
| Topic | 0.890 | 0.890 | **0.902** | 0.883 | 0.828 | 0.871 | 0.877 | 0.859 | 0.859 | 0.804 | 0.840 | 0.785 | 0.828 | 0.706 | 0.598 | 0.845 |
| Sentiment | 0.875 | **0.906** | 0.867 | 0.781 | 0.797 | 0.813 | 0.828 | 0.805 | 0.766 | 0.766 | 0.773 | 0.766 | 0.789 | 0.641 | 0.711 | 0.797 |
| Avg. | 0.847 | **0.859** | 0.859 | 0.810 | 0.753 | 0.835 | 0.828 | 0.751 | 0.796 | 0.698 | 0.750 | 0.719 | 0.780 | 0.675 | 0.605 | 0.776 |
| **Utility Constraint** | | | | | | | | | | | | | | | | |
| Target Language | **0.878** | 0.839 | 0.800 | 0.817 | 0.691 | 0.726 | 0.817 | 0.687 | 0.574 | 0.609 | 0.652 | 0.609 | 0.570 | 0.457 | 0.563 | 0.701 |
| Supportiveness | **0.848** | 0.808 | 0.808 | 0.808 | 0.702 | 0.801 | 0.788 | 0.702 | 0.709 | 0.636 | 0.649 | 0.623 | 0.709 | 0.649 | 0.563 | 0.728 |
| Consistency | 0.927 | 0.891 | **0.945** | 0.845 | 0.827 | 0.873 | 0.891 | 0.836 | 0.782 | 0.709 | 0.673 | 0.700 | 0.827 | 0.755 | 0.618 | 0.814 |
| Factuality | **0.758** | 0.757 | 0.711 | 0.724 | 0.600 | 0.714 | 0.725 | 0.614 | 0.642 | 0.528 | 0.571 | 0.486 | 0.578 | 0.566 | 0.468 | 0.636 |
| Helpfulness | **0.850** | 0.835 | 0.842 | 0.817 | 0.723 | 0.793 | 0.811 | 0.730 | 0.727 | 0.630 | 0.698 | 0.644 | 0.711 | 0.696 | 0.601 | 0.746 |
| Avg. | **0.830** | 0.814 | 0.804 | 0.792 | 0.689 | 0.769 | 0.789 | 0.697 | 0.692 | 0.603 | 0.655 | 0.600 | 0.667 | 0.653 | 0.551 | 0.714 |
| **Composition Type** | | | | | | | | | | | | | | | | |
| Chain | 0.725 | **0.732** | 0.718 | 0.693 | 0.568 | 0.664 | 0.674 | 0.605 | 0.566 | 0.463 | 0.537 | 0.489 | 0.551 | 0.538 | 0.444 | 0.606 |
| Selection | **0.822** | 0.785 | 0.782 | 0.785 | 0.646 | 0.742 | 0.798 | 0.709 | 0.701 | 0.595 | 0.683 | 0.607 | 0.672 | 0.666 | 0.567 | 0.709 |

Table 22: Detailed results of LLMs on diverse constraint dimensions and composition types. The highest performance overall is **bold**.

## I.3 Detailed Results of Each Task Type

Table 23 presents the DRFR of the selected LLMs for each task type. We find that the performance of LLMs across tasks is balanced overall. Relatively, LLMs perform better on tasks related to writing and role-playing, while they have shortcomings in Logical Reasoning, Advanced Chinese Understanding, and Open-ended Questions. All LLMs exhibit significant weaknesses in Logical Reasoning, which is consistent with the Reasoning Drawbacks found in AlignBench [6].

| Task Type | Fund. | Chi. | Open. | Prac. | Crea. | Pro. Writing | Cust. | Role. | Pro. Knowledge | Logic. | Overall |
|---|---|---|---|---|---|---|---|---|---|---|---|
| *Closed-Source Language Models* | | | | | | | | | | | |
| GPT-4-1106 | **0.783** | **0.751** | **0.761** | 0.810 | **0.845** | 0.808 | **0.870** | 0.856 | **0.838** | 0.681 | **0.800** |
| Claude-3-Opus | 0.752 | 0.729 | 0.722 | 0.805 | **0.845** | **0.816** | 0.864 | **0.874** | 0.722 | 0.698 | 0.788 |
| GLM-4 | 0.738 | 0.717 | 0.735 | **0.821** | 0.798 | 0.800 | 0.843 | 0.858 | 0.745 | 0.683 | 0.779 |
| ERNIEBot-4 | 0.732 | 0.721 | 0.680 | 0.802 | 0.804 | 0.759 | 0.828 | 0.824 | 0.757 | **0.718** | 0.764 |
| GPT-3.5-Turbo-1106 | 0.675 | 0.584 | 0.578 | 0.743 | 0.737 | 0.710 | 0.743 | 0.779 | 0.645 | 0.517 | 0.682 |
| *Open-Source Language Models* | | | | | | | | | | | |
| Qwen1.5-72B-Chat | 0.713 | 0.695 | 0.653 | 0.798 | 0.810 | 0.772 | 0.831 | 0.840 | 0.749 | 0.619 | 0.752 |
| Llama-3-70B-Instruct | 0.732 | 0.617 | 0.676 | 0.771 | 0.833 | 0.767 | 0.855 | 0.853 | 0.741 | 0.678 | 0.757 |
| InternLM2-20B-Chat | 0.641 | 0.595 | 0.619 | 0.713 | 0.751 | 0.676 | 0.778 | 0.792 | 0.691 | 0.512 | 0.678 |
| Qwen1.5-14B-Chat | 0.617 | 0.621 | 0.600 | 0.715 | 0.724 | 0.703 | 0.799 | 0.819 | 0.695 | 0.506 | 0.680 |
| Baichuan2-13B-Chat | 0.549 | 0.528 | 0.515 | 0.646 | 0.665 | 0.608 | 0.660 | 0.713 | 0.548 | 0.410 | 0.591 |
| Llama-3-8B-Instruct | 0.610 | 0.558 | 0.580 | 0.690 | 0.702 | 0.673 | 0.719 | 0.670 | 0.622 | 0.468 | 0.638 |
| Mistral-7B-Instruct | 0.530 | 0.394 | 0.578 | 0.647 | 0.686 | 0.604 | 0.713 | 0.686 | 0.494 | 0.457 | 0.592 |
| Qwen1.5-7B-Chat | 0.601 | 0.517 | 0.619 | 0.715 | 0.720 | 0.660 | 0.749 | 0.790 | 0.641 | 0.503 | 0.658 |
| InternLM2-7B-Chat | 0.628 | 0.517 | 0.553 | 0.712 | 0.622 | 0.662 | 0.692 | 0.743 | 0.598 | 0.479 | 0.634 |
| ChatGLM3-6B-Chat | 0.510 | 0.439 | 0.464 | 0.586 | 0.606 | 0.606 | 0.636 | 0.605 | 0.537 | 0.368 | 0.546 |

Table 23: Automated DRFR of LLMs on different task types. The highest performance among open-source models is underlined, while the highest performance overall is **bold**. "*Fund.*" denotes Fundamental Language Ability, "*Chi.*" denotes Advanced Chinese Understanding, "*Open.*" denotes Open-ended Questions, "*Prac.*" denotes Practical Writing, "*Crea.*" denotes Creative Writing, "*Pro. Writing*" denotes Professional Writing, "*Cust.*" denotes Custom Writing, "*Role.*" denotes Task-oriented Role Play, "*Pro. Knowledge*" denotes Professional Knowledge and "*Logic.*" denotes Logical Reasoning.

