# OpenReview forum: "Benchmarking Complex Instruction-Following with Multiple Constraints Composition"
_NeurIPS.cc/2024/Datasets_and_Benchmarks_Track — NeurIPS 2024 Track Datasets and Benchmarks Poster_

### Official Review · Reviewer_bTTo · 2024-06-22
**Extending instruction-following benchmark to a more complex and practical setup**

**Rating:** 7
**Confidence:** 3
**Clarity:** The paper is well-written and organiz…

**Review:**

Instruction following is a critical capability of LLMs in practice. The authors make a compelling case that existing benchmarks do not adequately capture the compositional nature of many real-world instructions.

To better model compositional nature, the authors define the taxonomy of constraints and compositions, which is well-motivated and seems comprehensive.

The dataset construction process appears thorough, with multiple stages of human annotation and validation.

The proposed RAL evaluation method leverages the strengths of both rule-based and LLM-based evaluation.

The experimental results provide valuable insights into the current capabilities and limitations of state-of-the-art LLMs in handling complex instructions. The analysis of performance across different constraint and composition types is particularly inspiring.

**Strengths:**

- Addresses an important gap in current LLM evaluation benchmarks with well-motivated and comprehensive taxonomy of instruction complexity
- High-quality manually constructed dataset
- Extensive experiments with both closed and open-source LLMs

**Additional Feedback:**

NA

**Correctness:**

The methodology appears sound and the results are presented clearly with appropriate statistical analysis

**Documentation:**

Dataset is provided in the attachment. Detailed description about how dataset is curate is provided in the paper.

**Ethics:**

No concern.

**Limitations:**

- The dataset is in Chinese, which is better to be bilingual or multilingal.
- While the dataset size (1,150 instructions) is reasonable, an even larger set could provide more statistical power.

**Opportunities For Improvement:**

- It would be interesting to see how to use this benchmark to guide post-training
- The authors could consider releasing a leaderboard to encourage ongoing benchmarking
- At line 156, "we analyze the distribution of composition types in real-world scenarios" -- what is the real-world scenario refer to? better to explain the source

**Relation To Prior Work:**

The authors provide a comprehensive overview of related work in instruction following evaluation and compositionality in NLP.

**Summary And Contributions:**

This paper introduces a new benchmark for evaluating large language models' (LLMs) ability to follow complex instructions with multiple constraints and compositional structures. Contributions include

- A hierarchical taxonomy of complex instructions is introduced, which is more comprehensive compared to existing works
- A manually constructed dataset of 1,150 complex instructions is proposed in the benchmark
- A novel evaluation method called Rule-Augmented LLM-based (RAL) evaluation is used to provide accurate scores

---

> ### Author Rebuttal · Authors · 2024-08-17
>
> Thanks for your comprehensive and detailed suggestions for our work! We hope our detailed response could address your concerns:
>
> > **Opportunities For Improvement 1:** It would be interesting to see how to use this benchmark to guide post-training
>
> Please refer to general response **4**.
>
> > **Opportunities For Improvement 2:** The authors could consider releasing a leaderboard to encourage ongoing benchmarking
>
> Thanks for your valuable suggestions! We are developing a leaderboard and will publish it soon, allowing new LLMs to join the benchmarking freely.
>
> > **Opportunities For Improvement 3:** At line 156, "we analyze the distribution of composition types in real-world scenarios" -- what is the real-world scenario refer to? better to explain the source
>
> Please refer to general response **3**.
>
> > **Limitations 1:** The dataset is in Chinese, which is better to be bilingual or multilingal.
>
> Please refer to general response **5**.
>
> > **Limitations 2:** While the dataset size (1,150 instructions) is reasonable, an even larger set could provide more statistical power.
>
> Thanks for your valuable suggestions! Since each instance of ComplexBench is carefully constructed and validated by humans, we choose the current data size to make it easier for strict quality control and human review. It's worth noting that the size of ComplexBench is significantly larger than previous instruction-following benchmark datasets, such as IFEval[1] (541 instructions), InfoBench[2] (500 instructions), and CELLO[3] (523 instructions). We will continue to enlarge ComplexBench in the future to obtain the evaluation results with more statistical power.
>
> [1] Jeffrey Zhou, Tianjian Lu, Swaroop Mishra, Siddhartha Brahma, Sujoy Basu, Yi Luan, Denny Zhou, and Le Hou. Instruction-following evaluation for large language models. arXiv preprint arXiv:2311.07911, 2023.
>
> [2] Yiwei Qin, Kaiqiang Song, Yebowen Hu, Wenlin Yao, Sangwoo Cho, Xiaoyang Wang, Xuansheng Wu, Fei Liu, Pengfei Liu, and Dong Yu. Infobench: Evaluating instruction following ability in large language models. Findings of ACL 2024.
>
> [3] Qianyu He, Jie Zeng, Wenhao Huang, Lina Chen, Jin Xiao, Qianxi He, Xunzhe Zhou, Jiaqing Liang, and Yanghua Xiao. Can large language models understand real-world complex instructions? In Proceedings of the AAAI Conference on Artificial Intelligence, volume 38, pages 18188–18196, 2024.

---

### Official Review · Reviewer_91Pj · 2024-07-24
**Comprehensive benchmark for complex instruction following**

**Rating:** 6
**Confidence:** 4
**Clarity:** The paper is well-structured and clea…

**Review:**

This paper presents a well-executed contribution to the field of LLM evaluation. As LLMs become increasingly capable and are applied to more complex real-world tasks, there is a growing need for benchmarks that can assess their ability to follow nuanced, multi-step instructions. ComplexBench addresses this need by providing a comprehensive structure framework for evaluating complex instruction following. I really like the RAL evaluation approach, which combines the strengths of rule-based and LLM-based methods. This approach evokes principles from traditional linguistics, bridging established wisdom with modern AI techniques. The Question-level Agreement study is also very important, as we are yet to know how good rule following is for LLMs to check the rules. However, a significant limitation is the exclusive use of Chinese text in the dataset. This mono-lingual focus substantially restricts the benchmark's generalizability, raising concerns about its applicability across different languages and cultural contexts.

**Strengths:**

1. Significance in the field: ComplexBench addresses a crucial gap in LLM evaluation by focusing on complex instruction following, which is increasingly important for real-world applications.
2. Research quality: The research is well-designed, with a comprehensive taxonomy, carefully constructed dataset, and novel evaluation method.
3. Bridging established wisdom with modern AI techniques: The RAL evaluation approach, which combines the strengths of rule-based and LLM-based methods, evokes principles from traditional linguistics
4. Agreement Validation: A critical aspect of using LLMs for task automation is assessing their alignment with human judgment. This paper conducts an extensive study on this front, demonstrating superior agreement between human evaluators and the RAL approach. This enhanced agreement validates the effectiveness of the RAL method and strengthens confidence in its evaluations.

**Additional Feedback:**

I would like to understand why only Chinese is used in the dataset. Also would different languages affect how you construct the dataset and apply constraints?

**Correctness:**

The authors provide detailed information about benchmark construction and evaluation, which seems correct.

**Documentation:**

Sufficent.

**Ethics:**

No major ethical concerns identified.

**Limitations:**

1. A key drawback of rule-based approaches is the potential artificiality of the generated data. Upon examining some data points in the dataset, I observed that certain samples deviate from realistic patterns. Some constraints or combinations present in the data are improbable in real-world scenarios (like extensive if-else pattern).

**Opportunities For Improvement:**

* Language diversity: Expanding the benchmark to include multiple languages would enhance its generalizability and applicability across different linguistic contexts. Having only Chinese in the dataset would greatly hurt the generality of the benchmark.
* In Figure 9, duplicated colors has been used for plotting, which is a bit confusing. Different colors or line style could be used instead.

**Relation To Prior Work:**

The work has clearly discussed its difference from previous works, especially in table 1.

**Summary And Contributions:**

This paper introduces ComplexBench, a new benchmark for evaluating llms' ability to follow complex instructions. The key contributions are:

1. A hierarchical taxonomy for complex instructions, including 4 constraint types, 19 constraint dimensions, and 4 composition types.
2. A manually constructed high-quality dataset of 1,150 instructions covering the proposed hierarchical taxonomy.
3. A novel evaluation method called Rule-Augmented LLM-based (RAL) evaluation that combines rule-based and LLM-based approaches.
4. Extensive experiments evaluating 15 LLMs on ComplexBench, revealing their strengths and weaknesses in following complex instructions.

---

> ### Author Rebuttal · Authors · 2024-08-17
>
> Thanks for your comprehensive and detailed suggestions for our work! We hope our detailed response could address your concerns:
>
> > **Opportunities For Improvement 1:** Language diversity: Expanding the benchmark to include multiple languages would enhance its generalizability and applicability across different linguistic contexts. Having only Chinese in the dataset would greatly hurt the generality of the benchmark.
>
> Please refer to general response **5**.
>
> > **Opportunities For Improvement 2:** In Figure 9, duplicated colors has been used for plotting, which is a bit confusing. Different colors or line style could be used instead.
>
> We appreciate your valuable feedback. We will use clearer and more distinct methods to present this figure to ensure better readability.
>
> > **Limitations:** A key drawback of rule-based approaches is the potential artificiality of the generated data. Upon examining some data points in the dataset, I observed that certain samples deviate from realistic patterns. Some constraints or combinations present in the data are improbable in real-world scenarios (like extensive if-else pattern).
>
> We highly value your concern about the alignment between our dataset and real-world scenarios. We believe ComplexBench is indeed aligned with real-world scenarios, supported by the following evidence:
>
> **Firstly**, our taxonomy of constraint and composition is established by integrating common constraints from previous controlled text generation tasks and analyzing instructions from real-world application scenarios. These constraint and composition types are widely present in real-world scenarios. The introduction of the rule-based method is intended to address the weakness of LLMs when evaluating a small portion (approximately 17%) of rule-verifiable, lexical, and format constraints. For other constraints that cannot be verified by rules, we directly use LLMs for evaluation. The choice of evaluation methods is determined after establishing the taxonomy of complex instructions, based on the characteristics of each constraint. Consequently, the rule-based methods do not introduce additional human prior knowledge during data construction.
>
> **Secondly**, during data construction, we collect reference instructions from real-world application scenarios and open-source instruction-following benchmarks and ask annotators to modify them manually to create high-quality evaluation data, ensuring that the instructions in ComplexBench are aligned with real-world scenarios as much as possible.
>
> **Lastly**, for instructions with complex constraint compositions (such as extensive if-else patterns), as shown in Figure 4 of our paper, these instructions are relatively rare in general scenarios but widely present in professional scenarios, such as task-oriented role play, information extraction, and knowledge base question answering. Therefore, these instructions of ComplexBench are constructed based on the complex task demands of real-world scenarios and have not deviated from realistic patterns.
>
> > **Additional Feedback:** I would like to understand why only Chinese is used in the dataset. Also, would different languages affect how you construct the dataset and apply constraints?
>
> Since the data we collected from real-world application scenarios is primarily in Chinese, we chose to construct ComplexBench in Chinese. We respectfully clarify that the taxonomy of constraint and composition in ComplexBench is language-agnostic and does not leverage features specific to Chinese, making it applicable to any language.

---

### Official Review · Reviewer_nFjW · 2024-07-25

**Rating:** 6
**Confidence:** 3
**Correctness:** Claims appear correct and dataset con…

**Review:**

**Quality:**
* Pros:
    * Clear, well-constructed taxonomy of constraints and compositions. The analysis of existing professional and general instructions in section 3.3 provides an excellent motivation/justification for the proposed taxonomy.
    * Dependency graph of constraints is intuitive for evaluation and also provides richer annotation for future use of the dataset.
    * Thorough evaluation of closed and open-source LLMs on ComplexBench, including testing out if decomposed instructions work well.
* Cons:
    * It would be interesting if there was a training split of ComplexBench so that one could study if LLMs can be trained to follow multiple constraints well.
    * In the chain constraint, it is mentioned that the output of $T_{k+1}$ is dependent on $T_1, \dots, T_k$. However, the example provided on Mona Lisa does not exhibit this dependence---the year of creation, background of creation, and impact of the work can all be independent of each other. If all chain samples are similar to this, then the sequential way of evaluating them seems odd.

**Clarity:**
* Cons:
     * The paper's clarity would be improved if running examples were provided. For instance, examples of existing benchmarks in the intro, an example of a reference instruction, task allocation, bias in the branching step in the data collection step, and an example/more description of the coherent test.
     * (minor) Can more information be provided on the general versus professional instructions? Is this open source? How were the constraints verified---through manual counting?

**Originality:**
* Pros: paper overcomes limitations of previous work, which focuses on simple compositions and naive evaluation of these compositions.

**Significance:**
* Pros: Paper tackles an important question in the community right now as interactions with LLMs become more complex.
* Cons: (minor) dataset is limited to Chinese, which may make it difficult for international researchers to know how to more deeply analyze and parse the data.

**Strengths:**

1. Clear, well-constructed taxonomy of constraints and compositions. The analysis of existing professional and general instructions in section 3.3 provides an excellent motivation/justification for the proposed taxonomy.
2. Dependency graph of constraints is intuitive for evaluation and also provides richer annotation for future use of the dataset.
3. Thorough evaluation of closed and open-source LLMs on ComplexBench, including testing out if decomposed instructions work well.
4. Paper tackles an important question in the community right now as interactions with LLMs become more complex.

**Additional Feedback:**

N/A

**Clarity:**

The paper's clarity would be improved if running examples were provided. For instance, examples of existing benchmarks in the intro, an example of a reference instruction, task allocation, bias in the branching step in the data collection step, and an example/more description of the coherent test.

**Documentation:**

Documentation is provided.

**Ethics:**

No ethical concerns.

**Limitations:**

Limitations are addressed.

**Opportunities For Improvement:**

General:
1. In the chain constraint, it is mentioned that the output of $T_{k+1}$ is dependent on $T_1, \dots, T_k$. However, the example provided on Mona Lisa does not exhibit this dependence---the year of creation, background of creation, and impact of the work can all be independent of each other. If all chain samples are similar to this, then the sequential way of evaluating them seems odd.
2. The paper's clarity would be improved if running examples were provided. For instance, examples of existing benchmarks in the intro, an example of a reference instruction, task allocation, bias in the branching step in the data collection step, and an example/more description of the coherent test.
3. Can more information be provided on the general versus professional instructions? Is this open source? How were the constraints verified---through manual counting?

For future work:
4. It would be interesting if there was a training split of ComplexBench so that one could study if LLMs can be trained to follow multiple constraints well.
5. The dataset is limited to Chinese, which may make it difficult for international researchers to know how to more deeply analyze and parse the data.

**Relation To Prior Work:**

Related work is discussed.

**Summary And Contributions:**

This paper study how to evaluate LLMs' ability to follow multiple dependent constraints. Previous benchmarks typically focus on a single constraint or naive compositions of constraints, such as the "and" constraint (e.g., "Produce a response that is in Spanish and under 100 words"). Moreover, they evaluate overall constraint satisfaction as an average across individual constraints. This paper instead presents a hierarchical taxonomy of constraint types and composition types, such as "and", a sequential "chain", a branching "selection", and nested combinations of these. They find that these constraints come up frequently in user interactions with LLMs, and construct a dataset ComplexBench consisting of samples whose instructions include multiple constraints in a dependency graph. For evaluation, the paper proposes to toggle between LLM and rule-based evaluators per constraint depending on if the constraint is hard (i.e., "numerical") or soft. After individual constraints are scored, overall evaluation of a response is based on the structure of the dependency graph, rather than naive averaging. They find that leading closed and open-source LLMs all struggle with multiple constraints, especially with more nesting and more complex constraints such as "chain" and "selection".

---

> ### Author Rebuttal · Authors · 2024-08-17
>
> Thanks for your comprehensive and detailed suggestions for our work! We hope our responses could address your concerns.
>
> > **Opportunities For Improvement 1:** In the chain constraint, it is mentioned that the output of $T_{k+1}$ is dependent on $T_1, T_2, …,T_k$. However, the example provided on Mona Lisa does not exhibit this dependence---the year of creation, background of creation, and impact of the work can all be independent of each other. If all chain samples are similar to this, then the sequential way of evaluating them seems odd.
>
> We respectfully clarify that there can be weak dependencies (consecutive tasks may be relatively independent of each other, as in the example on Mona Lisa) or strong dependencies (the outputs of all preceding tasks will affect the outputs of the subsequent tasks) between different tasks $T_1, T_2, …,T_k$ in the instructions containing Chain. Here, we also provide an example (translated into English) from our dataset where there are strong dependencies between different tasks:
>
> > We want to travel to Hawaii for 5 days and hope you can help us create a travel itinerary. For each day, please generate a daily plan, arranging 1-2 must-see attractions or activities. Then, recommend a few hotels or resorts for accommodation and specify the modes of transport between these locations. The language style should be relaxed and warm, and each day's plan should be no less than 400 words.
>
> Furthermore, we manually inspect all the instructions of ComplexBench containing *Chain* and count the number of instructions with weak or strong dependencies between different tasks. On the ComplexBench, out of 325 instructions of ComplexBench containing Chain, there are 229 instructions with strong dependencies between different tasks $T_1, T_2, …,T_k$, accounting for 70.5% of the total. This result demonstrates the soundness of our evaluation method.
>
> > **Opportunities For Improvement 2:** The paper's clarity would be improved if running examples were provided. For instance, examples of existing benchmarks in the intro, an example of a reference instruction, task allocation, bias in the branching step in the data collection step, and an example/more description of the coherent test.
>
> Thanks for your suggestions! We will make revisions in the following aspects:
>
> - **Examples of existing benchmarks**: We will add an example of existing benchmarks in the introduction.
> - **An example of human annotation (including reference instructions and task allocation)**: We will integrate detailed information about human annotation provided in the supplementary material (i.e., Section 5 of 'ComplexBench_Supplementary_Materials.pdf') into the appendix in our next paper revision and provide a specific example of data construction to facilitate a better understanding, as shown in general response **1**.
> - **Bias in the branching step**: We respectfully clarify that we have adopted **Selection Branch Expansion** to ensure the probability of the correct branch appearing at each position in Selection is equal, eliminating potential branching bias. To verify its necessity, we additionally analyze the distribution of correct branches in the instructions containing *Selection* before the **Selection Branch Expansion**. Specifically, we manually count the correct branch distribution of all selection compositions (one instruction may contain multiple selection compositions) in these data before **Selection Branch Expansion**, as shown in the table below:
>
> | Branch Number / Correct Branch | Branch 1 | Branch 2 | Branch 3 | Branch 4 |
> | ------------------------------ | -------- | -------- | -------- | -------- |
> | 2                              | 273      | 130      | -        | -        |
> | 3                              | 14       | 7        | 9        | -        |
> | 4                              | 21       | 19       | 21       | 20       |
>
> We found that annotators tend to select the first branch as the correct one, making Selection Branch Expansion necessary.
>
> - **More description of the coherent test**: We reorganize the description of the coherent test to ensure better readability. Please refer to the general response **2**.
>
> > **Opportunities For Improvement 3:** Can more information be provided on the general versus professional instructions? Is this open source? How were the constraints verified---through manual counting?
>
> Please refer to general response **3** for a detailed explanation of the real-world scenarios. In addition, these instructions are not open-source. And yes, we manually count the frequency of composition types.
>
> > **Opportunities For Improvement 4:** It would be interesting if there was a training split of ComplexBench so that one could study if LLMs can be trained to follow multiple constraints well.
>
> Please refer to general response **4**.
>
> > **Opportunities For Improvement 5:** The dataset is limited to Chinese, which may make it difficult for international researchers to know how to more deeply analyze and parse the data.
>
> Please refer to general response **5**.

---

### Official Review · Reviewer_Xp7s · 2024-07-25
**Rather positive review, some question related to dataset construction and license**

**Rating:** 6
**Confidence:** 4
**Correctness:** I have not seen any claims that are i…

**Review:**

I am rather positive about the proposed benchmark: the idea of generating more complex instruction-following benchmark is relevant and important for the NLP community. The human evaluation and newly proposed rule-based and LLM as a judge scoring procedure are also essential for understanding having a correct evaluation for the CompleBench task.

However, I have not completely understood the dataset construction procedure - maybe this can be detailed in the appendix using an example. More, there are many missing details about the annotators (both for dataset creation and human evaluation). At last, it should be useful to investigate whether there is a correlation between performance on existing datasets and some of the results on the ComplexBench scenario.

At the same time, I have not seen any details in the paper about licensing or distribution of the dataset.

**Strengths:**

* Providing a more complex instruction-following benchmark and dataset, by adding compositional constraints (such as sequence or branching).
* Good evaluation, including human annotators - that highlight that the proposed automatic rule+LLM evaluation metric has a better correlation with human evaluation
* The newly added evaluation combining rule-based and LLM as a judge which is relevant for ComplexBench

**Additional Feedback:**

None

**Clarity:**

In general the paper is well written.
However, there are a couple of parts which are still unclear: dataset creation (add an example), and also the "Coherent Test" in Section 5.2.3.

**Documentation:**

Maybe I am missing this part, but have not seen in the paper any details about licensing or distribution of the dataset.

**Ethics:**

No ethical concerns

**Limitations:**

I would have liked to see from the beginning that the benchmark is built for Chinese. There is no mention about this in the paper (first one is in Appendix A) and it was unclear why GLM and Ernie had scores close to OpenAI and Cohere models.

At the current moment, the missing details about the annotators is also an important limitation.

**Opportunities For Improvement:**

* Dataset construction procedure is unclear: the authors should provide an example in the appendix to understand the main steps and how the composed complex instructions are created. This is important in order not to have biases in the data.
* Missing details about the human annotation process: who were the annotators, were they trained, were they paid? This is very important both for dataset creation and human evaluation.
* Investigate whether there is a correlation between performance on existing datasets and some of the results on the ComplexBench scenario
* Missing details in the paper about licensing or distribution of the dataset.

**Relation To Prior Work:**

Prior work is well highlighted. However, I think that Topic-Following [1] is an example of Instruction-Following that also contains compositional structures (such as sequences or branching), but they are not treated and evaluated separately but rather in the context of a task-oriented dialogue agent that should respect all rules defining the task/scenario in natural language (including sequences of turns, what the bot should answer if the user asks about a certain topic and so on).

[1] - https://arxiv.org/abs/2404.03820

**Summary And Contributions:**

In the last couple of months, various benchmarks and datasets like IFEval, InFoBench, and CELLO proposed to investigate specific fine-grained instruction-following (or rule-following) abilities of LLMs. The current paper builds on top of these existing works by introducing a more complex instruction-following benchmark and dataset, called ComplexBench - the main difference is that it adds compositional constraints between simpler instructions and also a combined rule and LLM as a judge method for judging if an instruction is satisfied.

---

> ### Author Rebuttal · Authors · 2024-08-17
>
> Thanks for your comprehensive and detailed suggestions for our work! Here are our responses and improvements.
>
> > **Opportunities For Improvement 1:** Dataset construction procedure is unclear.
>
> We respectfully clarify that we have provided the guidelines for data construction in the supplementary material, i.e., Section 5 of 'ComplexBench_Supplementary_Materials.pdf'. Sections 5.1 and 5.2 present the specific requirements for the two key steps in data construction: **Data Annotation** and **Selection Branch Expansion** (as mentioned in Section 4.1). We have also provided a specific example of data construction to facilitate a better understanding in general response **1**.
>
> > **Opportunities For Improvement 2:** Missing details about the human annotation process.
>
> We respectfully clarify that we have provided information about the source and payment of annotators in the supplementary material, i.e., Section 5 of 'ComplexBench_Supplementary_Materials.pdf'. Specifically, we recruited 12 college students for data annotation of ComplexBench, and the total labor cost is approximately $2800.
>
> As for annotators training, all the annotators are required to complete a training tutorial that includes 50 samples. We provide feedback to help them calibrate the annotation criteria. In addition, during the annotation process, we conduct spot checks on annotated data. We also retrain the annotators based on the issues found in their annotations and require them to recheck such issues in their annotated data. The annotator training process for dataset creation and human evaluation is similar. We will incorporate these details in our revision.
>
> > **Opportunities For Improvement 3:** Investigate whether there is a correlation between performance on existing datasets and some of the results on the ComplexBench scenario
>
> Thanks for your valuable feedback. We compare the performance of representative LLMs across 3 prominent LLM evaluation benchmarks in addition to ComplexBench: IFEval, HumanEval, and MATH, focusing on instruction-following, coding, and mathematical ability, respectively. As shown in Table 1 of the following PDF, although the performance of various LLMs on ComplexBench is well correlated with their performance on other benchmarks, the rankings of LLMs on ComplexBench do not entirely correspond with those on the other three benchmarks. For instance, ChatGLM3-6B-Chat demonstrates outstanding coding and mathematical abilities among LLMs of similar scale, but it notably struggles with complex instruction-following. On the other hand, while Llama-3-70B-Instruct surpasses GPT-4-1106 on IFEval and ranks first, it still shows a performance gap with GPT-4-1106 on ComplexBench. This discrepancy is primarily in the areas of instructions with complex constraints composition, which are not covered by IFEval, indicating that ComplexBench can provide a complementary perspective for LLM evaluation.
>
> > **Opportunities For Improvement 4:** Missing details in the paper about licensing or distribution of the dataset.
>
> We respectfully clarify that we have provided the license of ComplexBench in the supplementary material, i.e., line 78 of 'ComplexBench_Supplementary_Materials.pdf'. ComplexBench is distributed under CC BY 4.0. The evaluation code of ComplexBench is distributed under the MIT license. This information will be highlighted in our revision.
>
> > **Limitations:** I would have liked to see from the beginning that the benchmark is built for Chinese. There is no mention about this in the paper (first one is in Appendix A) and it was unclear why GLM and Ernie had scores close to OpenAI and Cohere models.
>
> Thanks for your considerate suggestions! We will mention this in the introduction of our next paper revision, and release the English version of ComplexBench in the future to promote research on complex instruction-following across different languages.
>
> GLM and Ernie are known to be advanced closed-source models, especially in Chinese language tasks. Various existing works validate that their performance is close to SoTA LLMs (including OpenAI and Anthropic models). For example, [1] has shown that for constraint-following tasks, the performance of GLM-4 and Ernie is close to the SoTA model GPT-4o. [2] has shown that GLM-4 performs comparably to GPT-4-1106 and Claude-3-Opus across various tasks.
>
> > **Clarity:** However, there are a couple of parts which are still unclear: dataset creation (add an example), and also the "Coherent Test" in Section 5.2.3.
>
> As for dataset creation, please refer to the response to **Opportunities For Improvement 1**. As for the coherent test, we provide a refined description of this part in general response **2**, to ensure better readability.
>
> > **Relation To Prior Work:** Relation to Topic-Following.
>
> Thanks for your kind reminder. We believe the main difference between Topic-Following and our work mainly lies in two aspects:  Firstly, Topic-Following does not model the Chain composition of different constraints and the nested structures of various composition types.  In task-oriented dialogue tasks, the model needs to complete relatively independent tasks based on the user's instructions in each turn, rather than completing multiple interdependent tasks in sequence. Secondly, Topic-Following only considers a special case of the Selection composition that generates a standard template when the user mentions specific topics, without taking into account the diversity and complexity of selection conditions and branches. We will add this discussion to our revision.
>
> [1] Zhang, T., Shen, Y., Luo, W., Zhang, Y., Liang, H., Yang, F., ... & Zhou, Z. (2024). CFBench: A Comprehensive Constraints-Following Benchmark for LLMs. *arXiv preprint arXiv:2408.01122*.
>
> [2] GLM, T., Zeng, A., Xu, B., Wang, B., Zhang, C., Yin, D., ... & Wang, Z. (2024). ChatGLM: A Family of Large Language Models from GLM-130B to GLM-4 All Tools. *arXiv preprint arXiv:2406.12793*.

---

### Author Rebuttal · Authors · 2024-08-17

We sincerely appreciate the insightful comments from all the reviewers, which will surely make ComplexBench stronger!

As reviewers highlighted, we believe our paper tackles an important problem (**Reviewer Xp7s, Reviewer nFjW, Reviewer 91Pj, Reviewer bTTo**), introduces a well-motivated (**Reviewer bTTo**), comprehensive (**Reviewer 91Pj,**  **Reviewer bTTo**) and well-constructed (**Reviewer nFjW**) taxonomy of complex instructions and purposes a novel (**Reviewer 91Pj, Reviewer bTTo**) evaluation method. We appreciate that the reviewers find our paper is well-written (**Reviewer Xp7s, Reviewer bTTo**) and offer thorough (**Reviewer nFjW**) and extensive (**Reviewer 91Pj, Reviewer bTTo**) experiments.

In response to several shared concerns raised by the reviewers, we would like to provide the following clarifications.

1. **[Reviewer Xp7s, Reviewer nFjW] Provide an example of data construction to improve clarity.** We value this comment and provide a specific example (translated into English) about data annotation in Table 1 of the following PDF to facilitate a better understanding of the process. The objectives of the first two steps in data construction, i.e., **Reference Instruction Collection** and **Task Allocation** (as mentioned in Section 4.1) are to obtain **Reference Instruction** and **Task Requirements** fields of the table, respectively. Each field of the table corresponds to Table 2 of 'ComplexBench_Supplementary_Materials.pdf' in the supplementary material. We will integrate this example into the appendix in our revision.
2. **[Reviewer Xp7s, Reviewer nFjW] Make a clearer description of the coherent test.** Thanks for the valuable suggestion! We reorganize the description of this part with additional examples and formulas to improve readability.

> **The** **Coherent** **Test for Selection.** To comprehensively measure the performance of LLMs on different conditions of Selection, we merge the instructions with the same branches and selection functions but different conditions into the same task group. For example, the instruction about the Mona Lisa shown in Figure 1 and another instruction where everything else remains the same except the final condition "Painting: Mona Lisa" is changed to "Painting: Galloping horse" are merged into the same task group. The two instructions need to execute two different selection branches. We calculate the proportion of instructions with all scoring questions correct (Original Test) and group tasks with all scoring questions correct (Coherent Test). Formally, considering that there are $N$ instructions containing Selection, they are divided into $K$ task groups. Each instruction $i$ has $m_i$ scoring questions, and the result of the $j$-th scoring question is $r_{ij}^{'}$ (the same definition as Section 4.2). The results of Original Test (OT) and Coherent Test (CT) will be calculated as follows:
>
> $$ OT = \frac{1}{N} \sum_{i=1}^{N}(\bigwedge_{j=1}^{m_i}r_{ij}^{'}) $$
>
> $$ CT = \frac{1}{K}\sum_{k=1}^{K}\bigwedge_{i\in Group(k)}(\bigwedge_{j=1}^{m_i}r_{ij}^{'}) $$
>
> Instructions containing Selection are categorized as either single-layer or multi-layer nested, respectively. As shown in Figure 11, for single-layer Selection instructions, LLMs with stronger instruction-following abilities show a smaller performance drop in the coherent test, which better understands the selection structure. For more complex multi-layer nested Selection instructions, even the state-of-the-art LLM, GPT-4, achieves only 14.9% accuracy in the coherent test, while smaller-scale LLMs can’t perfectly follow any group of instructions. The results highlight current LLMs’ weaknesses in following multi-layer tree-structured instructions.

3. **[Reviewer nFjW, Reviewer bTTo] Provide more information about the distribution analysis of composition types in real-world scenarios.** We respectfully clarify that real-world scenarios refer to scenarios collected from an online LLM-based chat service platform. Our online chat platform serves more than a million users daily, ensuring the diversity of our collected samples. General instructions refer to the instructions used by individual users in routine scenarios, while professional instructions refer to the instructions used by enterprise-level users in business and research scenarios. We only use the portion of instructions that are granted for research purposes by users and conduct a strict deprivatization and desensitization process to protect user privacy.
4. **[Reviewer nFjW, Reviewer bTTo] Split training set of** **ComplexBench** **and use ComplexBench to guide post-training.** We respectfully clarify that ComplexBench is entirely manually constructed and has undergone rigorous quality control and human review. It is just used as a high-quality benchmark for testing and thus lacks enough data to split a substantial training set. We strongly agree with the importance of designing automated methods for training data augmentation to enhance the complex instruction-following capabilities of LLMs. Based on ComplexBench's taxonomy and evaluation methods, one possible implementation idea is as follows: We can leverage in-context learning and instruct LLMs to automatically add constraints to reference instructions and compose multiple instructions for creating instructions with complex constraint composition based on our taxonomy, and generate corresponding scoring questions. Then, we can select high-quality responses as the labels of training data based on the scoring results. We leave the improvement of the quality of automatically augmented instructions and scoring questions with minimal human involvement as important future work.
5. **[Reviewer nFjW, Reviewer 91Pj, Reviewer bTTo] Language diversity of ComplexBench.** Thanks for the insightful feedback! We promise that we will release the English version of ComplexBench in the future to promote research on complex instruction-following across different languages.

---

### Author Response · Authors · 2024-08-24
**Looking forward to your feedback**

Dear reviewers,

Thank you again for your insightful reviews. As the end of the discussion period draws near, we would like to ensure that we have adequately addressed all your concerns. If you have any further feedback, please do not hesitate to let us know.

---

### Decision · Program_Chairs · 2024-09-26

**Decision:**

Accept (Poster)

**Comment:**

The paper introduces ComplexBench, a new benchmark to evaluate large language models (LLMs) on following complex, multi-constraint instructions. It includes a comprehensive hierarchical taxonomy of constraints and a manually constructed dataset. The benchmark addresses gaps in current LLM evaluation methods, demonstrating strengths in assessing instruction complexity. Overall, the benchmark provides valuable insights into LLM capabilities and limitations in handling complex instructions.